# The Chemical Fluctuation Theorem governing gene expression

Seong Jun Park [1,2,3], Sanggeun Song [1,2,3], Gil-Suk Yang [1], Philip M. Kim [4],
Sangwoon Yoon [2], Ji-Hyun Kim [1] & Jaeyoung Sung [1,2,3]

Gene expression is a complex stochastic process composed of numerous enzymatic reactions with rates coupled to hidden cell-state variables. Despite advances in single-cell technologies, the lack of a theory accurately describing the gene expression process has restricted a robust, quantitative understanding of gene expression variability among cells. Here we present the Chemical Fluctuation Theorem (CFT), providing an accurate relationship between the environment-coupled chemical dynamics of gene expression and gene expression variability. Combined with a general, accurate model of environment-coupled transcription processes, the CFT provides a unified explanation of mRNA variability for various experimental systems. From this analysis, we construct a quantitative model of transcription dynamics enabling analytic predictions for the dependence of mRNA noise on the mRNA lifetime distribution, confirmed against stochastic simulation. This work suggests promising new directions for quantitative investigation into cellular control over biological functions by making complex dynamics of intracellular reactions accessible to rigorous mathematical deductions.

[1] Creative Research Initiative Center for Chemical Dynamics in Living Cells, Chung-Ang University, Seoul 06974, Korea. [2] Department of Chemistry, Chung-Ang University, Seoul 06974, Korea. [3] National Institute of Innovative Functional Imaging, Chung-Ang University, Seoul 06974, Korea. [4] Terrence Donnelly Center for Cellular and Biomolecular Research, Department of Molecular Genetics and Department of Computer Science, University of Toronto, Toronto M5S 3E1 ON, Canada. Correspondence and requests for materials should be addressed to S.Y. (email: sangwoon@cau.ac.kr) or to J.-H.K. (email: jihyunkim@cau.ac.kr) or to J.S. (email: jaeyoung@cau.ac.kr)

Every chemical reaction is a stochastic process, and every life form is operated by chemical reactions catalyzed by enzymes. Enzyme activity varies strongly even among clonal enzymes because of its coupling to enzyme conformation and environmental variables[1–3]. This is evidenced by the fact that the time elapsed during a single enzymatic turnover and the product number were found to be stochastic variables with far greater randomness than predicted by conventional enzyme kinetics, which assumes a constant enzyme activity[4–6]. An important question to ask here is, in what way and how accurately do life forms achieve the order required to develop and sustain their lives from the disordered reaction events of single enzymes. This question was first addressed in the context of gene expression because of its fundamental importance in biology[7–10].

Modern single-molecule experiments have shown that the cellular control over gene expression is imperfect; even among cells carrying the same exact gene, the abundance of proteins produced by gene expression was found to vary from cell to cell[8,9], causing phenotype variations[11–13]. From a chemical physics perspective, the fluctuation in protein levels among genetically identical cells is dependent on the chemical dynamics of gene expression, which, in turn, depends on the mechanism of gene expression and the environment-coupled dynamics of the elementary processes constituting gene expression. Thus, the variability in the protein levels can be adjusted through various experimental measures that effectively change the dynamics of transcription and translation, the two major chemical processes constituting gene expression[14–16]. Recently, the number of such experimental studies has grown rapidly[17,18]. However, a robust, quantitative understanding of the chemical dynamics of intracellular gene networks and their relationship to gene expression variability is still lacking.

The reason being, it is challenging to construct a rigorous model for the chemical dynamics of gene expression, which is composed of multi-channel or multi-step reactions with rates coupled to cell-state variables. Examples of the cell-state variables coupled to the gene expression rate include the populations of RNA polymerase (RNAP) and ribosomes[19]; the populations of transcription factors and micro-RNAs[20]; the interaction strength of genes with RNAP and transcription factors[21]; the gene copy number[22,23]; the phase of the cell cycle[24]; the density of nutrients[25]; and the conformation of chromosomes[26]. All of these cell-state variables are stochastic variables that differ from cell to cell and fluctuate over time. The stochastic dynamics of the entire cell state and its influence on the chemical dynamics of gene expression are much too involved to be accurately described by the conventional kinetic network model or any specific mathematical model[27,28]. Typically, in quantitative studies in this field, one cell-state variable is chosen as the control variable, and the dependence of the gene expression statistics on this control variable is analyzed using the conventional kinetic network model. However, a successful quantitative explanation of experimental data is extremely rare in this field, because the conventional kinetic network model cannot effectively account for the interaction of the gene network with the rest of the cell-state variables, or the environmental variables. Recently, to describe intracellular networks interacting with hidden cell environments, a new model and stochastic kinetics have been developed[29], yet this theory is still in its adolescence.

In experimental research, on the other hand, impressive advances have recently been made; development of single-molecule fluorescence in situ hybridization[30,31], multiplexed error-robust FISH[32], and single-cell RNA sequencing techniques[33–35] have enabled measurements of cell-to-cell variation in messenger RNA (mRNA) levels for single-gene expression systems[23,36] and for the entire genome of cells[33,35]. Studies on the single-gene expression system clearly show that cell-to-cell variation in mRNA levels depends on the molecular mechanism of transcription[23,36]. Meanwhile, a genome-wide expression statistics revealed a global trend in the relationship between the variance and the mean of mRNA levels in Escherichia coli (E. coli)[37]. This finding raised the intriguing question as to whether a universal law governing the expression variability of every gene exists[38]. However, whether such a universal law exists or whether a unified, quantitative understanding of the above-mentioned experiments is even possible remains unknown.

Here we show that a simple equation that governs cell-to-cell variation in gene-expression levels among a clonal population of cells does exist, and it holds for any gene-expression system. We call this equation the Chemical Fluctuation Theorem (CFT). The CFT provides the exact relationship between the fluctuation in the number of product molecules and the environment-coupled dynamics of product creation and annihilation processes. Combined with an accurate transcription model that takes into account the non-Poisson transcription dynamics and the influence from cell environments in a collective and complete manner, the CFT provides a unified, quantitative explanation for cell-to-cell variability in the mRNA number for various experimental systems. Using both the CFT and the quantitative model of transcription dynamics optimized from our analysis, we make quantitative predictions for the dependence of the mRNA noise on the mRNA lifetime fluctuation. According to our predictions, the mRNA noise increases with the cell-to-cell heterogeneity in mRNA lifetime, but interestingly, the mRNA noise decreases with increasing fluctuation in mRNA lifetime caused by non-Poisson mRNA degradation dynamics in each cell. The correctness of these predictions is confirmed against stochastic simulation.

In the analysis of experimental data, we find that the mean mRNA level dependence of non-Poisson mRNA noise, or the difference between the relative variance and the inverse mean of the mRNA level, is far more sensitive to the transcription dynamics than the Fano factor or other previously used measures. From the analysis, we find that the transcription dynamics is more sophisticated than that assumed in previous studies. The gene-state switching process makes an important contribution to non-Poisson mRNA noise[39,40]; yet, it alone cannot explain experimental data for the mRNA level dependence of mRNA noise. To quantitatively explain these data, we utilize new transcription models in which transcription of activated genes is a non-Poisson process. Fast-growing E. coli cells show an interesting oscillation in the time-correlation function (TCF) of the active gene transcription rate and have far less mRNA noise than their slowly growing counterparts with an exponentially decaying TCF. The growth condition-dependent TCF can be understood in terms of the reaction dynamics of the transcription process. The oscillation period in the TCF of the transcription rate progressively increases with time, which cannot be explained with the conventional assumption that the transcription dynamics is the same across the cell population or with use of Cox's renewal theory[41].

By analyzing lacZ gene expression under various constitutive promoters in E. coli[23], we find that the RNAP binding affinity of constitutive promoters also fluctuates with a rate of about 100 Hz or greater. This finding is consistent with the experimental data reported in refs. [42,43] (see Supplementary Note 1). This is much faster than the repressor-regulated gene-state switching, which occurs with a rate of about 2 Hz or less in the case of strong promoters. RNAP level fluctuation is found to be a major source of the environment-induced correlation between the transcription levels of different gene copies.

For the successful application of the CFT to quantitative analyses of various transcription systems with different control

**a**

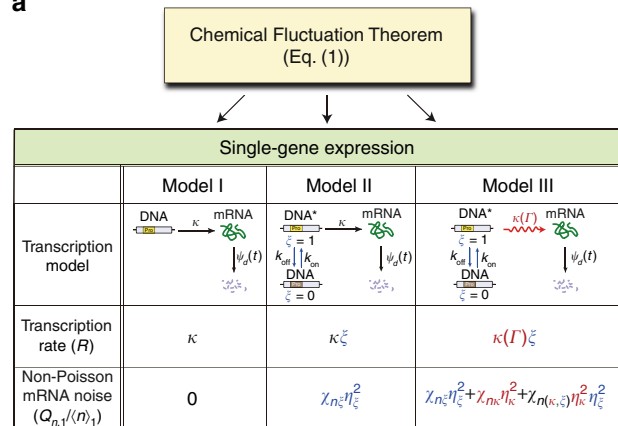

**b**

| Multi-gene expression | | |
|---|---|---|
| Transcription rate ($R = \sum_{i=1}^{g} R_i$) $g$: Random variable | $g\kappa$ | $\kappa \sum_{i=1}^{g} \xi_i$ | $\sum_{i=1}^{g} \kappa_i(\Gamma)\xi_i$ |
| Non-Poisson mRNA noise ($Q_n/\langle n \rangle$) | $\eta_g^2$ | $\dfrac{1}{\langle g \rangle} \dfrac{Q_{n,1}}{\langle n \rangle_1} + \eta_g^2$ | $\dfrac{1}{\langle g \rangle} \dfrac{Q_{n,1}}{\langle n \rangle_1} + \eta_g^2 + \dfrac{\langle g(g-1)\rangle}{\langle g \rangle^2} C_n$ |

**Fig. 1** Chemical Fluctuation Theorem (CFT) applied to various transcription models. Equation (1) relates the variance of mRNA number to the TCF of the transcription rate, whose mathematical form depends on the transcription network model. **a** In Model I, the transcription rate is given by constant $\kappa$. In Model II, the gene state switches stochastically between the active state ($\xi = 1$) and the inactive state ($\xi = 0$), for which the transcription rate is given by $\kappa \xi$, with $\kappa$ being a constant and $\xi$ being a stochastic variable. In Model III, the transcription rate is given by $\kappa(\Gamma)\xi$, where $\kappa(\Gamma)$ is a stochastic variable dependent on the cell state $\Gamma$. For each transcription model, the CFT yields the variance in the mRNA number. Non-Poisson mRNA noise for single-gene transcription is defined by $\left(\sigma_{n,1}/\langle n \rangle_1\right)^2 - \langle n \rangle^{-1}$, where $\sigma_{n,1}^2$ and $\langle n \rangle_1$ denote the variance and mean of the number of mRNA produced by a single-gene copy, denoted by $Q_{n,1}/\langle n \rangle_1$ in the main text ($Q_{n,1}$ denotes $\sigma_{n,1}^2/\langle n \rangle_1 - 1$). The duration time of each repressed and unrepressed gene state is reported to be an exponentially distributed random variable[36,56]; the mean is, respectively, denoted by $k_{on}^{-1}$ and $k_{off}^{-1}$. For mRNA lifetimes, we assume an arbitrary distribution, $\psi_d(t)$. The analytic expression of the non-Poisson mRNA noise obtained from Eq. (1) is tabulated for each model (Supplementary Methods). $\eta_q^2$ denotes the relative variance, $\langle \delta q^2 \rangle / \langle q^2 \rangle$, of variable $q$, ($q \in \{\kappa, \xi\}$). The respective susceptibilities $\chi_{n\xi}$, $\chi_{n\kappa}$, and $\chi_{n(\kappa,\xi)}$ of the mRNA noise to $\eta_\xi^2$, $\eta_\kappa^2$, and $\eta_\xi^2 \eta_\kappa^2$ are determined by TCFs of the fluctuations in transcription rate factors, $\kappa$ and $\xi$, and the survival probability of mRNA (see the text below Eq. (2)). **b** Non-Poisson mRNA noise among cells with multiple gene copies. The gene copy number, $g$, is a stochastic variable. For this system, the non-Poisson mRNA noise is defined by $(\sigma_n/\langle n \rangle)^2 - \langle n \rangle^{-1} (\equiv Q_n/\langle n \rangle)$. $Q_n$ denotes $\sigma_n^2/\langle n \rangle - 1$, with $\sigma_n^2$ and $\langle n \rangle$ being the variance and mean, respectively, in the number of mRNA copies across the cells with gene copy number variation. For all the three models, Model I–III, additional non-Poisson mRNA noise emerges from the gene copy number variation. $\eta_g^2$ denotes the relative variance $\sigma_g^2/\langle g \rangle^2$ in the gene copy number. For Model III, the non-Poisson mRNA noise among cells with gene copy number variation also emerges from the environment-induced correlation between the transcription levels of different gene copies. $C_n$ denotes the mean-scaled correlation between the number $n_1$ of mRNAs produced by the first gene copy and the number $n_j$ of mRNA produced by another gene copy, e.g., the $j$-th, i.e., $C_n = \langle \delta n_1 \delta n_j \rangle / \langle n_1 \rangle \langle n_j \rangle$ ($j \neq 1$)

variables, it is essential to construct an accurate model for the transcription rate coupled not only to the control variable, but also to uncontrolled or hidden cell-state variables. Below, for various experimental systems, we demonstrate our new method

to construct a quantitative model for the transcription rate coupled to hidden environmental variables, for which we lack the *a priori* information required to construct an explicit and accurate model.

## Results

**Chemical Fluctuation Theorem.** Transcription consists of several major chemical processes, including the binding of RNAP to the promoter, the activation of the RNAP–promoter complex, and transcriptional elongation during which mRNA is actually synthesized. Each of these processes, in turn, consists of a number of elementary reactions among biomolecules, whose conformation and reactivity fluctuate under the influence of cell environments. The transcription rate is not a constant but, rather, a stochastic variable, whose dynamical properties depend on the microscopic details of transcription and its coupling to cell environments. For genes under transcriptional regulation mechanisms, the fluctuation in the transcription rate becomes even more pronounced. It was recently shown that the transcription rate fluctuation caused by cell-to-cell variation in the RNAP level is an important source of cell-to-cell variability in the abundance of highly expressed proteins in *E. coli*[23,44], and that there exists a simple mathematical relationship between the first two moments of both the RNAP and the protein levels[29,44].

This suggests the existence of a general relationship between the fluctuation in the transcription rate, $R$, and cell-to-cell variability in the copy number, $n$, of mRNA, which we find here as

$$\sigma_n^2(t) = \langle n(t) \rangle + \int_0^t d\tau_1 \int_0^t d\tau_2 \, S(t-\tau_1) S(t-\tau_2) \langle \delta R(\tau_1) \delta R(\tau_2) \rangle. \tag{1}$$

Equation (1) shows that the variance, $\sigma_n^2$, of the mRNA number deviates from the mean, $\langle n(t) \rangle [= \int_0^t dt \langle R(\tau) \rangle S(t-\tau)]$, in the presence of the transcription rate fluctuation, $\delta R(t) = R(t) - \langle R(t) \rangle$; furthermore, this deviation is completely determined by the TCF of the transcription rate fluctuation and the survival probability, $S(t)$, of mRNA, or the probability that mRNA created at time 0 has not suffered an annihilation as of time $t$. The TCF of the transcription rate fluctuation vanishes only when transcription is a Poisson process with a constant or deterministic rate, and its precise definition is given in Supplementary Methods. Through this definition, the TCF of the product creation rate fluctuation can be related to the microscopic dynamics of the product creation reaction. As long as product decay is a renewal process[41], Eq. (1) holds exactly, irrespective of the time profile of the survival probability and the stochastic properties of the transcription process, which may be dependent on the transcription mechanism, cell environments, and the product number in the presence of feedback regulations. Equation (1) is applicable to a broad class of biological networks repeatedly creating product molecules, including translation as well as transcription (Supplementary Note 2). When product creation is a stationary process and the product lifetime distribution is a simple exponential function, Eq. (1) reduces to the result in ref. [29]. In addition, this equation correctly reduces to other previously reported results in the corresponding limits[45–49] (Supplementary Note 3).

Equation (1) cannot be easily derived from the chemical master equation or its variations when product decay is a general non-Poisson process, for which the decay rate of each product molecule at a given time differs from product to product depending on its survival time. In Supplementary Methods, we present two separate derivations of Eq. (1) and connect it with

well-established laws in probability theory, a transient version of Little's law, and the law of total variation[50].

It is possible to generalize the CFT to the case where the mRNA lifetime distribution is strongly heterogeneous among cells (Supplementary Note 4). The generalized CFT is exploited to investigate the effects of cell-to-cell heterogeneity in the mRNA lifetime later in this work. Throughout this work, we assume that the mRNA degradation process is not strongly correlated with the transcription process.

Equation (1) shows that the dynamics of the environment-coupled chemical processes constituting transcription are collectively manifested in cell-to-cell variation of the mRNA level through the TCF of the overall transcription rate. In experiments, the dependence of the variance on the mean in mRNA levels can be adjusted by changing, for example, the gene regulation mechanism and strength, the promoter-RNAP interaction strength, the gene copy number, and the cell-to-cell distribution of RNAP levels, because each of these cell-state variables effectively changes the TCF of the transcription rate. Therefore, an accurate analysis of cell-to-cell variation in mRNA level requires the correct relationship between the TCF of the transcription rate and the control variables in the experiment. To find the correct relationship between the two, we need to have an accurate mathematical description of the transcription rate that is coupled not only to the control variable but also to all the remaining cell-state variables.

**Vibrant network model for single-gene transcription**. Two popular kinetic models of transcription are the simple Poisson process with a constant rate (Model I)[51] and a network of Poisson transition processes between two gene states, each with its own transcription rate constant (Model II)[23,36,39]. There also exists a more complex transcription model involving a greater number of gene states[52]. This model is in better agreement with experimental results; however, it is uncertain whether this model accurately represents actual transcription dynamics, because transcription at each gene state can be a non-Poisson, multi-step enzyme process whose rate is coupled to various cell-state variables. Despite the development of several new approaches (Supplementary Note 5[53–55]), quantitatively accounting for the influence of complex cell environments on the transcription rate remains challenging within the framework of the conventional kinetic model or any other specific mathematical model.

To solve this problem, we take a different approach here, where a specific and explicit description is used only for the control variable-dependent part of the transcription rate, while, for the environmental variable-dependent part, a general and formal description is used[29]. For example, if a gene-state switching rate is experimentally controlled, but the rate of the ensuing transcription process is not, we use Model III in Fig. 1. Note that, in Model III, the controllable gene-state switching process is modeled explicitly, whereas uncontrolled transcription following gene-activation is modeled implicitly, meaning that the dependence of the uncontrolled transcription rate, $\kappa(\Gamma)$, on hidden cell-state variable, $\Gamma$, is unspecified. The cell-state dependent transcription process of the unrepressed gene is represented by the wavy arrow in Model III. The wavy arrow stands for the vibrant reaction process, whose rate is a stochastic variable dependent on the cell-state including the system. In comparison, the plain arrow represents a reaction process with a constant rate coefficient independent of cell-state. As we demonstrate below, the CFT enables us to obtain the general analytic result for the fluctuation in the number of product molecules created by an intracellular network involving vibrant reaction processes, for which we may not have a priori knowledge.

There are gene expression systems where the waiting time distributions of the repressed and unrepressed gene states are approximately given by exponential distributions[36,56]. For these systems, we can model gene activation and deactivation as Poisson processes with constant rates, $k_{on}$ and $k_{off}$ (Fig. 1). However, no experimental evidence supports a Poisson transcription process of the activated gene, and little quantitative information is available regarding environment-coupled dynamics of activated gene transcription, making it difficult to represent the transcription process in terms of a fully explicit model. We circumvent this problem by using the concept of a vibrant reaction process, which can represent any type of multi-step, multi-channel reaction with rate coupled to environmental variables.

Let us first apply Eq. (1) to Model III. The overall transcription rate of Model III can be written as $R = \xi\kappa(\Gamma)$, where $\xi$ is a stochastic variable representing a gene state, with the value being 0 for the repressed gene state and 1 for the unrepressed gene state, and $\kappa(\Gamma)$ represents the transcription rate of the unrepressed gene, which depends on cell-state variable $\Gamma$. For Model III, Eq. (1) yields

$$\eta_{n,1}^2 = \frac{1}{\langle n \rangle_1} + \chi_{n\kappa}\eta_\kappa^2 + \chi_{n\xi}\eta_\xi^2 + \chi_{n(\kappa,\xi)}\eta_\kappa^2\eta_\xi^2, \tag{2}$$

in the steady state (Supplementary Methods). $\eta_{n,1}^2$ and $\langle n \rangle_1$ denote the relative variance and the mean number of mRNA created by a single-gene copy in the steady state, respectively. The mean mRNA number $\langle n \rangle_1$ is given by $\langle n \rangle_1 = \langle \xi \rangle \langle n \rangle_{1.\max}$, where $\langle \xi \rangle$ and $\langle n \rangle_{1,\max}$ denote the probability of the unrepressed gene state, $\langle \xi \rangle = k_{on}/(k_{on} + k_{off})$ and the maximum value, $\langle n \rangle_{1,\max} = \langle \kappa \rangle \tau_m$, of the mean mRNA number for the case with $\langle \xi \rangle = 1$. $\tau_m$ denotes the mean mRNA lifetime, i.e., $\tau_m = \int_0^\infty dt\, S(t)$. Throughout this work, $\eta_q^2$ denotes the relative variance or noise of $q$, i.e., $\eta_q^2 = \langle \delta q^2 \rangle / \langle q^2 \rangle$ ($q \in \{\kappa, \xi\}$). Equation (2) shows that $\eta_{n,1}^2$ deviates from $\langle n \rangle_1^{-1}$ because of fluctuation in $\xi$ and $\kappa$, the two factors of the transcription rate, and the deviation is bilinearly proportional to $\eta_\xi^2$ and $\eta_\kappa^2$. The susceptibility or response of the mRNA noise to fluctuation in one of the two rate factors is determined by its TCF, $\phi_q$: $\chi_{nq} = \tau_m^{-2}\int_0^\infty dt_1\int_0^\infty dt_2 S(t_1)S(t_2)\phi_q(|t_1 - t_2|)$ ($q \in \{\kappa, \xi\}$). Similarly, the bilinear susceptibility $\chi_{n(\kappa,\xi)}$ is given by $\chi_{n(\kappa,\xi)} = \tau_m^{-2}\int_0^\infty dt_1\int_0^\infty dt_2 S(t_1)S(t_2)\phi_\kappa(|t_1 - t_2|)\phi_\xi(|t_1 - t_2|)$. For Model III, we have $\phi_\xi(t) = e^{-t(k_{on}+k_{off})}$ and $\eta_\xi^2 = k_{off}/k_{on}$ (Supplementary Note 6). In contrast, $\phi_\kappa(t)$ and $\eta_\kappa^2$ are unknown at this moment; however, information about them can be extracted from experimental data, as shown later in this work.

For Models I and II, Eq. (1) yields $\eta_{n,1}^2 = \langle n \rangle_1^{-1}$ and $\eta_{n,1}^2 = \langle n \rangle_1^{-1} + \chi_{n\xi}\eta_\xi^2$, respectively, which can also be obtained from Eq. (2) by setting $\eta_\kappa^2 = \eta_\xi^2 = 0$ for Model I and setting $\eta_\kappa^2 = 0$ for Model II.

Equation (2) is exact as long as $\xi$ and $\kappa$ are independent of each other. It is applicable not only to Model III, but also to other models in which $\xi$ and $\kappa$ bear different meanings and stochastic properties compared to those in Model III. This will be exploited in the analysis of various experiments in this work.

Equation (2) tells us that the effects of the cell-state dynamics and its coupling to the rate, $\kappa(\Gamma)$, of the vibrant reaction process are manifested in product noise through the TCF of $\kappa(\Gamma)$. However, product noise is not sensitive to other microscopic details of the environmental dynamics and its coupling to the reaction rate, so there can be numerous environment-coupled network models that yield the same product noise. It is remarkable that, even if the two rate factors, $\xi$ and $\kappa$, are independent of each other, the product noise is contributed from

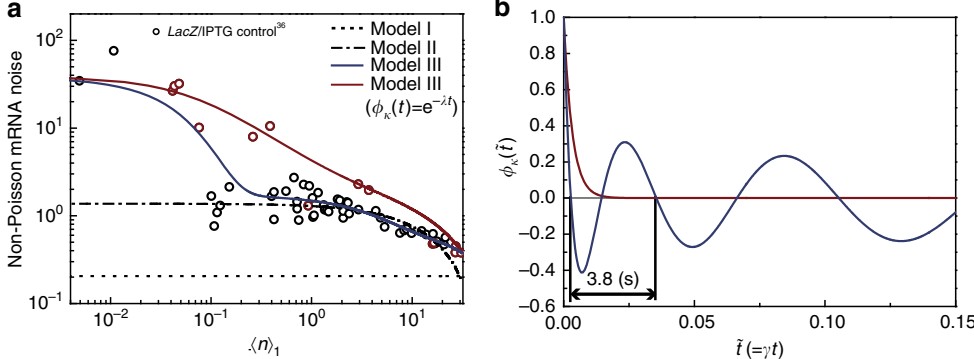

**Fig. 2** Dynamics of transcription rate fluctuation consistent with cell-to-cell variation in the number of mRNA expressed from repressor-regulated *lacZ* in *E. coli*. **a** (circles) Experimental data for the mean mRNA number $\langle n \rangle_1$ per single-gene copy and non-Poisson mRNA noise $Q_n/\langle n \rangle_1 (= \langle g \rangle Q_n/\langle n \rangle)$ for various concentrations of inducer IPTG[36]. The decay rate, $\gamma$, of the *lacZ* mRNA is 1/120 Hz[36]. The red circles represent the data obtained from slowly growing cells with doubling times greater than 45 min. The experimental data were reported in Fig. 3 of ref. [36]. (dotted line) The result of Model I, for which $Q_n/\langle n \rangle_1$ is the same as the Fano factor ($F_g \cong 0.21$) of the gene copy number variation (Supplementary Note 16). (dot-dash line) Result of Model II. Non-Poisson mRNA noise also emerges from the gene-state switching process. (blue solid line) Result of Model III. The fluctuation in transcription rate $\kappa$ produces additional mRNA noise. By comparing Model III and the entire data, we extract the time profile of the TCF, $\phi_\kappa(t)$, of transcription rate $\kappa$. (red line) Result of Model III with $\phi_\kappa(t)$ modeled as $\exp(-\lambda t)$, which is in good agreement with only the red circle data. The optimized value of $\lambda/\gamma$ is 306. The major contributors to non-Poisson mRNA noise are those from the gene-state switching process and its bilinear coupling with the ensuing transcription process, corresponding to the last two terms on the R.H.S. of Eq. (2) (Supplementary Note 17 and Supplementary Fig. 16a). **b** (blue line) $\phi_\kappa(t)$ extracted from the entire data shown in **a** using Model III. (red line) Exponential TCF, $\phi_\kappa(t) = e^{-\lambda t}$, extracted from the red circle data in **a**, obtained from the slowly growing cells. The dependence of the non-Poisson mRNA noise on the mean mRNA level is consistent with the known mechanism of IPTG-regulated transcription (see Supplementary Note 6)

the bilinear coupling term, the last term on the right-hand-side (R.H.S.) of Eq. (2), which makes it possible to extract information about the dynamics of $\kappa$ from the dependence of the product noise on $\xi$. Due to the bilinear coupling term, the mRNA noise in Eq. (2) is not given by the simple sum of the mRNA noise originating from the control variable-dependent part of the transcription network and the mRNA noise originating from the environmental variable-dependent part. This means that these two mRNA noises do not designate the "intrinsic" and "extrinsic" noise terms that often appear in the literature[8,28,57] (Supplementary Note 7).

**Effects of gene copy number variation**. Phillips and co-workers recently showed that gene copy number variation is an important source of mRNA noise while investigating the dependence of mRNA variability on the promoter architecture[23]. Here, we extend Eq. (2) to account for the effects of gene copy number variation. We once again start from Eq. (1). Let us consider cells with $g$ identical copies of the system gene. Because the total number of transcription events in any time interval is the sum of the number of transcription events for each gene copy in the interval, the total transcription rate $R$ is given by $R = \sum_{i=1}^{g} R_i$, with $R_i$ being the transcription rate from the $i$-th copy of the gene. The number, $g$, of gene copies is a stochastic variable with a fluctuation time scale much longer than the time scale of the individual transcription process.

The mean mRNA number in the multi-gene system is simply given by $\langle n \rangle = \langle g \rangle \langle n \rangle_1$, with $\langle g \rangle$ and $\langle n \rangle_1$ being the mean gene copy number and the mean number of mRNA produced per gene copy, respectively. The mRNA noise in the multi-gene system is contributed from gene copy number variation as well as from the transcription dynamics of individual genes. The transcription dynamics of individual genes manifests itself in the non-Poisson noise, $\eta_n^2 - \langle n \rangle^{-1} (\equiv Q_n/\langle n \rangle)$, much more evidently than it does on conventional measures of mRNA level fluctuation, such as the variance, the Fano factor, and the relative variance of the mRNA level (Supplementary Note 8).

For the multi-gene transcription system of Model III, Eq. (1) yields

$$\frac{Q_n}{\langle n \rangle} = \frac{1}{\langle g \rangle} \frac{Q_{n,1}}{\langle n \rangle_1} + \eta_g^2 + \frac{\langle g(g-1) \rangle}{\langle g \rangle^2} C_n, \tag{3}$$

where $\eta_g^2$ and $C_n$ denote the relative variance of the gene copy number and the mean-scaled correlation between the mRNA levels produced by different copies of a single-gene, defined by $C_n = \langle \delta n_i \delta n_j \rangle / \langle n_i \rangle \langle n_j \rangle$ $(i \neq j)$, respectively, (Supplementary Methods). Values of $\langle g \rangle$ and $\eta_g^2$ can be estimated by using either the static model[23] or the dynamic model[57,58] of replication (Supplementary Note 9). The correlation between the transcription levels of different gene copies emerges from fluctuations in global environmental variables such as the cellular levels of RNAP, transcription factors including sigma factors, and nutrients[8,22,29].

Equation (3) clearly shows that non-Poisson mRNA noise is contributed from the gene copy number noise and the correlation between the transcription levels of different gene copies. However, for Model III, these contributions are independent of the mean mRNA level (Supplementary Note 10), so the mean mRNA level-dependent change in non-Poisson mRNA noise results only from the first term on the R.H.S. of Eq. (3), non-Poisson mRNA noise produced by the single-gene transcription, which carries information about the environment-coupled transcription dynamics of individual genes.

The analytic result for non-Poisson mRNA noise is given in Fig. 1 for Models I–III. $C_n$ vanishes for Models I and II (Supplementary Note 6), but not for Model III due to environment-induced correlations between the transcription rates of different gene copies. $C_n$ cannot be greater than the non-Poisson noise, $Q_{n,1}/\langle n \rangle_1$, produced by a single gene, but the two quantities have the same order of magnitude when the major part of gene expression variability originates from fluctuation in the common environment shared by the gene copies[29].

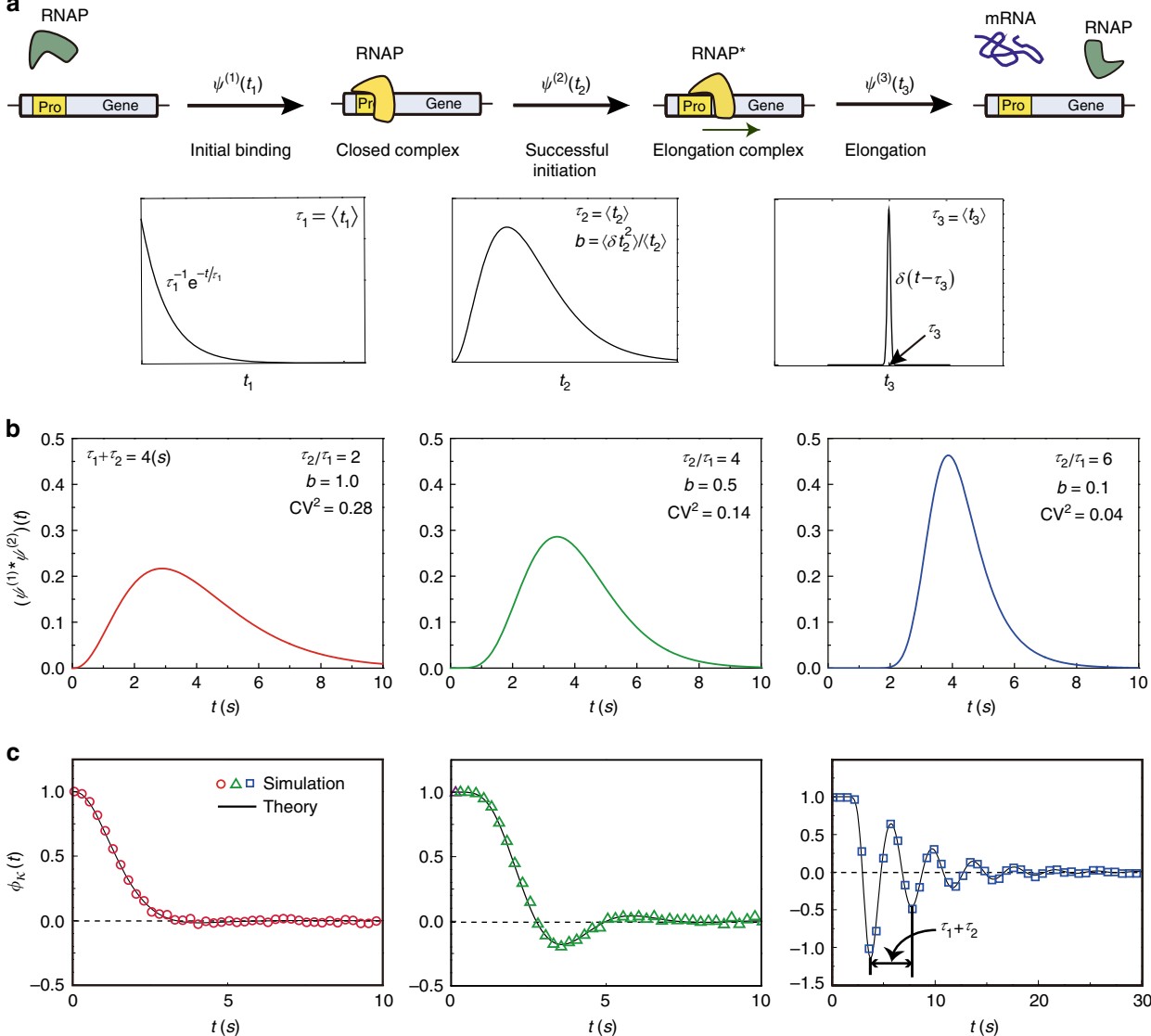

**Fig. 3** Non-Poisson transcription dynamics and TCF of the transcription rate of an unrepressed gene. **a** A simple model of the non-Poisson transcription process: (Step 1: Initial binding) initial binding of RNAP to promoter to form the closed complex; (Step 2: Successful initiation) transition of the RNAP-promoter complex into the elongation complex; and (Step 3: Elongation) synthesis of mRNA. Step 1 is modeled as a Poisson process with the mean reaction time, $\tau_1$, for which the reaction waiting time is distributed according to $\psi^{(1)}(t_1) = \tau_1^{-1}e^{-t_1/\tau_1}$. The reaction waiting time of Step 2 is modeled as a gamma distribution, $\psi^{(2)}(t_2) = t_2^{a-1}e^{-t_2/b}/(\Gamma(a)b^a)$, with $\langle t_2 \rangle = ab(\equiv \tau_2)$ and $\langle \delta t_2^2 \rangle/\langle t_2 \rangle \equiv b$. The elongation process (Step 3) is a highly sub-Poisson process for which the reaction time is modeled as $\psi^{(3)}(t_3) = \delta(t_3 - \tau_3)$. The next round of RNAP binding to the promoter is not allowed before the preceding RNAP completes the second step to leave the promoter. During transcriptional elongation by RNAPs, other RNAP can associate with the promoter and proceed to the next step. Multiple RNAPs can simultaneously perform the elongation process. **b** Distribution of transcription waiting times or the times between successive transcription events of the single-active gene copy. This distribution is given by the convolution of $\psi^{(1)}(t)$ and $\psi^{(2)}(t)$. The shape of the transcription waiting time distribution is shown for three different sets of parameter values for $\tau_1$, $\tau_2$, and $b$. The mean value, $\tau_1 + \tau_2$, is fixed at 4 s. **c** TCF, $\phi_\kappa(t)$, of the active gene transcription rate corresponding to each transcription waiting time distribution (see Supplementary Note 18). When the coefficient of variation (CV) in the transcription waiting time is small enough, $\phi_\kappa(t)$ exhibits an oscillatory feature. For the model shown in **a** every cell has the same transcription dynamics, in which case the oscillation period in $\phi_\kappa(t)$ is constant in time and approaches the mean transcription waiting time, $\tau_1 + \tau_2$, as the CV of the transcription waiting time decreases. In the presence of coupling to a disordered environment, the period of the oscillation in $\phi_\kappa(t)$ gradually increases over time, as shown by the blue line in Fig. 2b (Supplementary Note 6 and Supplementary Figure 7). Further details on the stochastic simulation method used in Fig. 3b, c can be found in Supplementary Note 19

**Transcription statistics of *lacZ* gene controlled by IPTG.** Golding and co-workers[36] investigated cell-to-cell variation in the number of mRNA expressed from the *lacZ* gene in *E. coli*, changing the transcription level by controlling the concentration of isopropyl β-D-1-thiogalactopyranoside (IPTG), which induces transcription by inhibiting the binding of the *lac* repressor to the gene. These experimental data have not yet been analyzed with

variation in the gene copy number taken into account, which is done here with use of Model I–III. For this system, the mRNA lifetime distribution is approximately an exponential function;[36] accordingly, we model mRNA degradation as a Poisson process with a constant rate, $\gamma$, $(\psi_d(t) = -\partial S(t)/\partial t = \gamma e^{-\gamma t}$ in Fig. 1) in the analysis of this system. As shown in Fig. 2a, we achieve an excellent quantitative explanation of these experimental data by

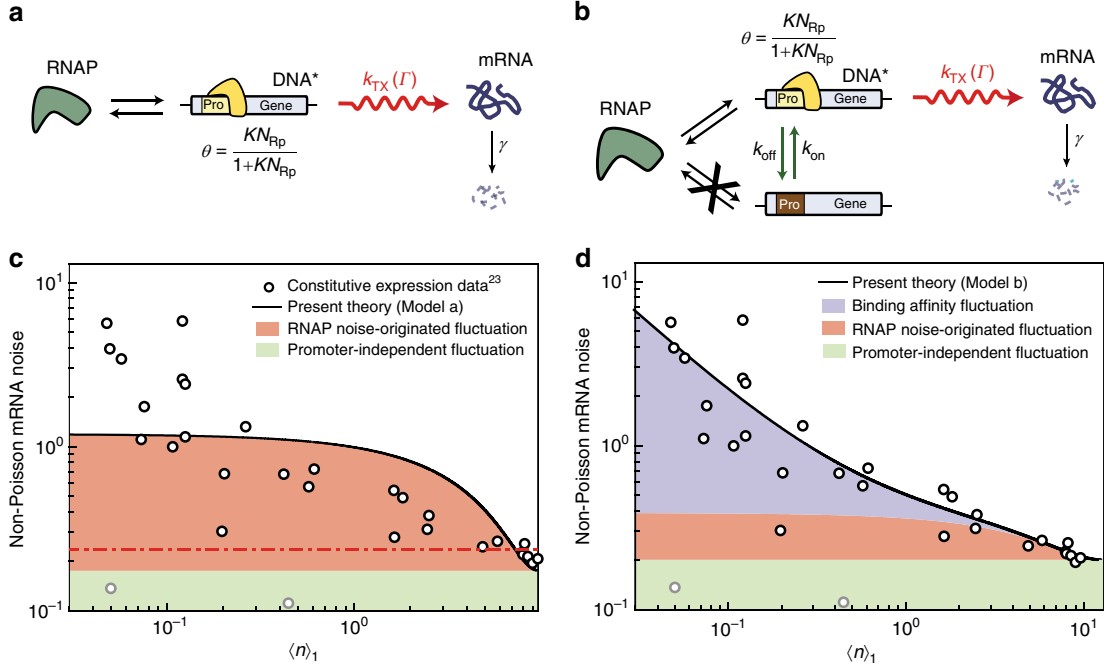

**Fig. 4** Effects of RNAP-promoter binding affinity fluctuation on non-Poisson mRNA noise in constitutive expression. **a** Transcription model of a constitutive gene without promoter strength fluctuation. The overall transcription rate of a single-gene copy is modeled by $R_1 = \left[ KN_{Rp}/(1 + KN_{Rp}) \right] k_{TX}(\Gamma)$, where the number $N_{Rp}$ of RNAP is a stochastic variable, but the RNAP binding affinity $K$ of the promoter is a constant. **b** Transcription model of constitutive gene with promoter strength fluctuation due to conformational dynamics of DNA. The overall transcription rate is given by the same formula as transcription model **a**, but with $K$ being a dichotomous stochastic variable whose value takes either 0 or $K_0$. Our analysis shows fluctuation in the value of $K_0$ is negligible (Supplementary Note 20). **c** (circles) Experimental results for the mean number $\langle n \rangle_1$ of mRNA per gene copy and non-Poisson noise $Q_n/\langle n \rangle_1 (= \langle g \rangle Q_n/\langle n \rangle)$ from the constitutive expression data reported in Fig. 2 of ref. [23]. (red dot-dash) Prediction of the previous model proposed in ref. [23]. (solid line) Result of transcription model **a** best fitted to the experimental data. **d** (solid line) Result of transcription model **b** best fitted to the experimental data. Binding affinity fluctuation in the constitutive expression occurs much faster than in the repressor-regulated gene expression (Supplementary Figs. 2 and 10). For constitutive gene expression, transcription model **b** provides a better quantitative explanation of the experimental data than Model III. However, in the absence of fluctuation in $N_{Rp}$, transcription model **b** becomes equivalent to Model III (Supplementary Fig. 12)

using Model III and the assumption that IPTG changes the rate, $k_{off}$, at which the *lac* repressor changes the gene state from the unrepressed to the repressed state (Supplementary Note 6), consistent with the known mechanism of IPTG and the conclusion of ref. [36].

The success of Model III attests that transcription rate, $\kappa$, of the unrepressed gene is not a constant, as assumed in Model II, but a dynamic stochastic variable. From the quantitative analysis of the experimental data with Model III, we can extract the time profile of the TCF, $\phi_\kappa(t)$, of $\kappa$, which exhibits an oscillatory feature (Fig. 2b). The experimental results shown in Fig. 2 cannot be explained by assuming a model with a monotonically decaying TCF or a white noise model of transcription rate fluctuation (Supplementary Figures 2 and 4). The oscillatory $\phi_\kappa(t)$ can be understood from the known mechanism of transcription, only if the distribution of transcription waiting times, or the times between successive transcription events of the unrepressed gene, has a peak with a small relative fluctuation (Fig. 3)[59].

To understand the oscillatory TCF, we consider the well-known mechanism of activated gene transcription, shown in Fig. 3a, which is represented by the wavy arrow in Model III. For the mechanism, the oscillation in $\phi_\kappa(t)$ emerges when the successful initiation of the open RNAP-promoter complex, consisting of a number of consecutive chemical processes, is the slow rate-determining step. When the initial approach and binding of RNAP to the promoter, which has greater randomness than the ensuing transcription processes, get slow, $\phi_\kappa(t)$ becomes

a monotonically decaying function (Fig. 3b, c), which is the case for slowly growing cells with less RNAP.

The progressively increasing oscillation period of $\phi_\kappa(t)$ shown in Fig. 2b signifies that the transcription dynamics is heterogeneous among a clonal population of cells (Supplementary Note 6 and Supplementary Fig. 7), which cannot be explained by assuming that every cell has the same transcription dynamics or that transcription is a renewal process with a single waiting time distribution independent of cell environments (Supplementary Fig. 21).

The oscillatory TCF of the transcription rate was previously observed for the human cell system[60], but it results from a different origin, that is, the oscillatory dynamics of chromatin remodeling. The period of this oscillatory dynamics is order of 10 h and far greater than the oscillation period of the TCF of the *E. coli* transcription rate, which is about 3.8 s (Fig. 2b), and close to the mean time between successive transcription events.

In actual bacterial transcription, the rate coefficient of the bimolecular association between RNAP and the promoter may not be constant either, because of the RNAP binding affinity fluctuation of the promoter associated with conformational dynamics of DNA. This has an important consequence for cell-to-cell variation in the number of mRNA copies created by constitutive gene expression, as we show in later in this work.

Under the alternative assumption that IPTG changes $k_{on}$ rather than $k_{off}$, neither Model II nor III provides a satisfactory explanation of the experimental data shown in Fig. 2 (Supplementary Note 6 and Supplementary Fig. 2c).

**Promoter strength-dependent transcription statistics**. Phillips and co-workers[23] recently showed that the relationship between the variance and mean in the mRNA level depends on the molecular mechanism of transcription and the experimental control variable. In their quantitative analysis, the authors first addressed the effects of both gene copy number variation and fluctuation in the RNAP number, $N_{Rp}$, on mRNA level variation. However, the authors' analysis was based on a simple model of transcription dynamics, in which a single gene transcription is a simple Poisson process with the rate linearly proportional to $N_{Rp}$, and the environment-induced correlation between the transcription levels of different gene copies is negligible. For this

simple model, Eq. (3) yields $Q_n/\langle n \rangle = \langle g \rangle^{-1}\eta_{N_{Rp}}^2 + \eta_g^2$, independent of the mean mRNA number (Supplementary Note 11), inconsistent with the experimental results (Fig. 4c, d).

Using Eq. (1) and the models shown in Fig. 4a, b, we reanalyze these experimental results in Fig. 4, obtained for the *lacZ* gene under various constitutive promoters[23]. We assume that the *lacZ* mRNA in this system also shows the same exponential decay as the *lacZ* mRNA in the system investigated in ref. [36]. Because the RNAP binding affinity, $K$, of the promoter is the experimental control variable, the single-gene transcription rate is modeled as $R_1 = \left[KN_{Rp}/(1 + KN_{Rp})\right]k_{TX}(\Gamma)$. Again, the control variable-dependent part of the transcription rate is modeled explicitly,

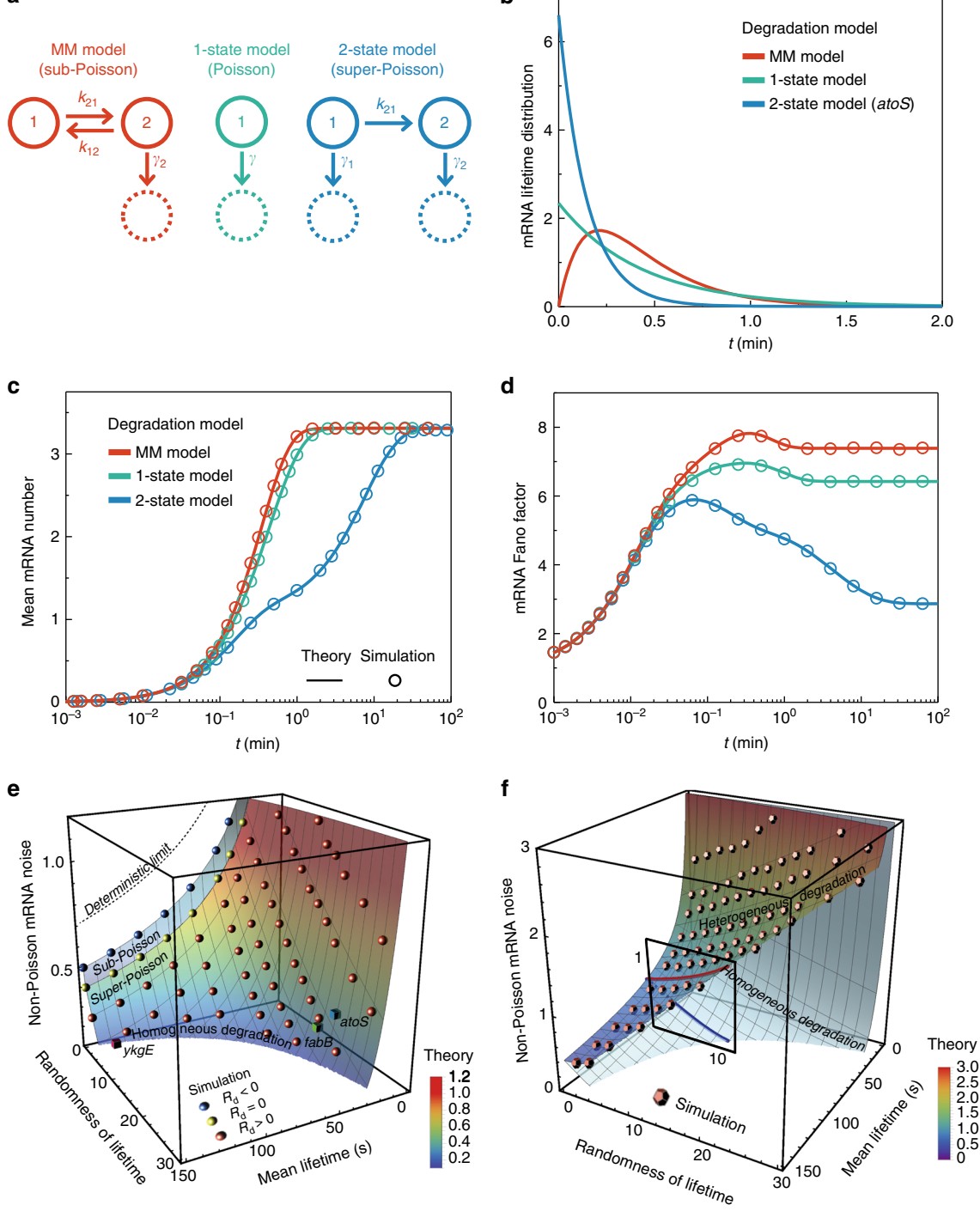

whereas the environment-dependent part, $k_{TX}(\Gamma)$, is modeled implicitly. In the model shown in Fig. 4a, $k_{TX}$ and $N_{Rp}$ are stochastic variables, but the RNAP binding affinity, $K$, of the promoter is not. In contrast, in the model shown in Fig. 4b, $K$ is also a stochastic variable accounting for fluctuations in the RNAP binding affinity of the promoter caused by the conformational dynamics of DNA. If the *lacZ* gene was expressed under promoters subject to a feedback regulation, $K$ would have been dependent on the product number as well (Supplementary Note 12).

RNAP number, $N_{Rp}$, is a global environmental variable that affects the transcription rates of all genes in the same manner, so cell-to-cell fluctuation in $N_{Rp}$ is an important source of the correlation, $C_n$, between the gene-expression levels of different genes or gene copies (Supplementary Note 13)[46]. According to Eq. (3), the global environment-induced correlation contributes to non-Poisson mRNA noise. For the models considered in Fig. 4, $N_{Rp}$ is nonlinearly coupled to the control variable, $K$, in the control variable dependent part, $\theta$, of the transcription rate as $\theta = KN_{Rp}/(1 + KN_{Rp})$, which makes $C_n$ dependent on the control variable $K$, i.e., $C_n \cong \eta^2_{N_{Rp}}/(1 + \overline{KN}_{Rp})^2$ with over-bar designating the mean (Supplementary Note 14 and Supplementary Fig. 11).

The model shown in Fig. 4a provides a better explanation of the experimental data than the model proposed in ref. [23], but its prediction is still qualitatively different from the data. However, we achieve an excellent quantitative explanation of the experimental results using the model in Fig. 4b, showing that, even for constitutive expression, the RNAP binding affinity, $K$, of the promoter is, in fact, not a constant, but a stochastic variable, attributable to the conformational dynamics of DNA[56,61] (Fig. 4d and Supplementary Note 15).

**mRNA noise dependency on the mRNA lifetime distribution.** There are many cell systems where the mRNA lifetime distribution is a non-exponential function for which the master equation or other existing methods cannot provide an accurate description. However, as demonstrated here, the CFT provides an accurate description of the mRNA noise for any mRNA lifetime distribution, when each mRNA degradation process is a renewal process.

The CFT makes it clear that when transcription is a Poisson process, mRNA noise becomes Poisson noise, i.e., $\sigma^2_n(t) = \langle n(t) \rangle$, regardless of the mRNA lifetime distribution. This is consistent with ref. [62], where the authors show that mRNA noise is independent of the number of reaction steps composing mRNA degradation given that transcription is a Poisson process.

However, the CFT also makes it obvious that mRNA noise is dependent on the mRNA lifetime distribution whenever transcription is a non-Poisson process with non-vanishing TCF. For example, we compare the mRNA noise predicted by the CFT and Model III for three different models of mRNA degradation in Fig. 5a: the sub-Poisson Michaelis–Menten process, the one-state Poisson process, and the two-state super-Poisson process[63–65]. Each represents a renewal process that is characterized by the mRNA lifetime distribution shown in Fig. 5b, and the mRNA lifetime distribution varies depending on the model in question. In this comparison, we set the mean mRNA lifetime the same for all three models and use the transcription part of Model III optimized by our analysis of the experimental data shown in Fig. 2 for the slowly growing *E.coli* system.

As shown in Fig. 5c, the steady-state mean mRNA number saturates to the same steady-state value regardless of mRNA lifetime fluctuation. However, non-Poisson mRNA noise varies between the models even in the steady state. The mRNA noise is found to be smallest for the two-state model of the super-Poisson mRNA degradation dynamics, but greatest for the sub-Poisson mRNA degradation model. These theoretical predictions are confirmed to be correct, in agreement with stochastic simulation (Fig. 5c, d), suggesting that mRNA noise decreases with increasing mRNA lifetime fluctuation caused by non-Poisson mRNA degradation dynamics.

For example, the mRNA of *atoS*, *fabB*, and *ykgE* in *E. coli* reportedly show bi-exponential lifetime distributions[66], and degradation of these mRNA were modeled by the 2-state super-Poisson process shown in Fig. 5a[63–65]. As shown in Fig. 5e, for this model, both the CFT and stochastic simulation tell us that non-Poisson mRNA noise decreases as the mRNA lifetime fluctuation increases. This and other previous models of mRNA degradation assume that mRNA degradation dynamics is not influenced by heterogeneous cell environments.

When mRNA degradation dynamics is strongly coupled to heterogeneous cell environments, the mRNA lifetime distribution differs from cell-to-cell and the overall mRNA degradation process is a non-renewal process to which the CFT is not applicable; however, we generalize the CFT to encompass this case in Supplementary Note 4. The generalized CFT indicates that mRNA noise increases with cell-to-cell heterogeneity in the mRNA degradation dynamics or lifetime distribution. For example, in Fig. 5f, we present the prediction of the generalized

**Fig. 5** Quantitative prediction for mRNA noise dependence on mRNA lifetime fluctuation. **a** Models of the mRNA degradation process: (left) the Michaelis–Menten sub-Poisson process; (middle) one-state Poisson process; (right) two-state super-Poisson process. mRNA degradation starts from state 1. The three models represent renewal processes with different distributions of mRNA degradation time. Randomness, $R_d$, in mRNA degradation time or lifetime is defined by the relative variance of mRNA lifetime minus unity. For sub-Poisson mRNA degradation, $R_d < 0$; for a Poisson process, $R_d = 0$; and for a super-Poisson process, $R_d > 0$. **b** mRNA lifetime distributions. (red line) Michaelis–Menten (MM) sub-Poisson process; (green) 1-state Poisson process; (blue line) 2-state super-Poisson process (Supplementary Note 16). The 2-state super-Poisson model has been used to model the bi-exponential lifetime distribution of *atoS*, *fabB*, and *ykgE* mRNA[63–65]. The mean mRNA lifetime is set equal to 25.8 s, the mean lifetime of *atoS* mRNA. **c, d** Time dependence of the mean and Fano factor in the number of mRNA transcribed by a single gene. The mean and Fano factor of mRNA number are calculated from equation (M1-6) and Eq. (1) for Model III, optimized from our analysis in Fig. 2 for experimental data obtained from slowly growing *E. coli*[36] (see Supplementary Table 1). The fraction of the active gene state is set to 1/2. **e, f** Dependence of the steady-state non-Poisson mRNA noise on the mean and randomness of the mRNA lifetime for two models: for the two-state super-Poisson mRNA degradation model without cell-to-cell heterogeneity (**e**) and for the 1-state Poisson mRNA degradation model, but with cell-to-cell heterogeneity in the rate (**f**). The theoretical prediction is made by the CFT in **e**, but by the generalized CFT, equation (M8-5) in **f**. (surface) Theoretical prediction. (spheres) Simulation results. (cubes) Prediction for *atoS*, *fabB*, and *ykgE* mRNA transcribed under IPTG inducible *lac* promoter in slowly growing *E. coli* shown in Fig. 2, given the 2-state super-Poisson model of mRNA degradation is valid[66]. Non-Poisson mRNA noise is measured by $\Delta(x) [\equiv Q_n/\langle n \rangle_1 - Q_n/\langle n \rangle_1 |_{x=1}]$, or the mean mRNA number dependent component of non-Poisson mRNA noise produced by single-gene transcription (see Supplementary Note 6). Both models yield the same bi-exponential mRNA lifetime distribution. mRNA noise increases with the randomness in the mRNA lifetime originating from cell-to-cell heterogeneity in the mRNA degradation rate, but decreases with an increase in the randomness of mRNA lifetime caused by non-Poisson mRNA degradation dynamics (Supplementary Note 4 and Supplementary Movies 1 and 2)

CFT for Model III, but with the bi-exponential mRNA lifetime distribution resulting from cell-to-cell heterogeneity in the mRNA degradation rate. In this model, the bi-exponential mRNA lifetime distribution is contributed from two cell groups each with their own differing exponential mRNA lifetime distribution. According to the prediction of the generalized CFT, non-Poisson mRNA noise increases with the mRNA lifetime fluctuation originating from cell-to-cell heterogeneity in the mRNA degradation dynamics. It is noteworthy that, even though these two models have the same bi-exponential mRNA lifetime distribution, the homogeneous mRNA degradation model considered in Fig. 5e and the heterogeneous mRNA degradation model considered in Fig. 5f show opposing trends in the dependence of mRNA noise on mRNA lifetime fluctuation.

This result suggests that the dependence of mRNA noise on mRNA lifetime fluctuation can serve as a probe of the main contributor to the non-exponential mRNA lifetime distribution. When cell-to-cell heterogeneity in the mRNA degradation dynamics is the main contributor, mRNA noise increases with mRNA lifetime fluctuation. On the other hand, when it is the homogeneous non-Poisson mRNA degradation dynamics, mRNA noise decreases as mRNA lifetime fluctuation increases.

## Discussion

Equation (1) tells us that non-Poisson mRNA noise emerges from fluctuations in the transcription rate[22,57]. According to our analysis shown in Fig. 2a, transcriptional regulation by the gene-state switching process inevitably increases transcription rate fluctuation and, consequently, non-Poisson mRNA noise as well. The rate of the ensuing transcription process of the RNAP-promoter complex also suffers large fluctuations; however, the dynamics of the process is a strongly non-Poisson, non-renewal process with small randomness, producing unexpectedly low mRNA noise (see Supplementary Note 6). This result shows that E. coli does not always exploit diversity in phenotype. It is equipped with special transcription dynamics of the gene in the active state to produce negligible mRNA noise compared to mRNA noise inevitably produced in the gene-state switching step required for adaptation or environment-dependent gene expression (Supplementary Fig. 16).

It was recently observed that the variance in the protein level was quadratically proportional to the mean[67]. Similarly, the variance, $\sigma_n^2$, in the mRNA level appears to be a quadratic function of the mean mRNA number, $\langle n \rangle$ (Supplementary Fig. 9). Our analysis shows that the origin of the quadratic dependence of $\sigma_n^2$ on $\langle n \rangle$ is the gene-copy number variation and the correlation between the transcription levels of different genes, which are independent of the control variable or the transcription dynamics of individual genes (see Supplementary Note 8). In contrast, non-Poisson mRNA noise originating from these control variable-independent sources, which corresponds to the last two terms on the R.H.S. of Eq. (3), is constant in the mean mRNA level. Therefore, the mean mRNA level dependence of the non-Poisson mRNA, resulting from the first term on the R.H.S. of Eq. (3), is far more sensitive to the single-gene transcription dynamics, which is exploited in our analysis shown in Fig. 2.

The non-Poisson mRNA noise data in Fig. 4 for the constitutive promoter are smaller than the data for the promoter regulated by repressors (Fig. 2a), even when the mean mRNA level and the magnitude of the transcription rate fluctuation are similar for both. This is because the dynamics of the gene-state switching caused by conformational dynamics of the constitutive promoters occurs faster than the dynamics associated with the reversible binding of repressors to promoters (Supplementary Fig. 1); according to the CFT, mRNA noise decreases as the speed of the transcription rate fluctuation increases. In Saccharomyces

cerevisiae as well, non-Poisson mRNA noise for constitutive genes was smaller than the noise for genes under additional regulation mechanisms[68].

The mathematical form of the CFT is in no way affected by the presence of feedback regulation, cell-to-cell communication, or other types of complication in the product creation process. A practical application of the CFT to the quantitative analysis of the reaction networks comprising more complicated processes is another interesting topic we leave for future research. However, we briefly discuss the application of the CFT to gene expression networks with feedback regulation in Supplementary Note 12, where we note that the TCF of the protein number is crucial information for such an application. Our derivation can be extended to obtain the TCF of the protein number in a straightforward manner, which is to appear elsewhere.

We present the CFT, arguably the first mathematical equation that chemical fluctuation in living cells actually obeys. Combined with our new type of transcription models, the CFT provides a unified, quantitative explanation of cell-to-cell variation in mRNA number for various experimental systems. Using a quantitative transcription model developed in our analysis, we make quantitative predictions for the dependence of mRNA noise on the mRNA lifetime distribution. This work proposes a promising, new approach to quantitative investigations into stochastic chemical dynamics of intracellular networks interacting with cell environments, marking an advance that would have been unobtainable through the master equation and other existing approaches. Examples of reaction networks composed of non-Poisson processes can be found in a vast variety of topics in both natural and social science, to which the CFT can be applied for quantitative fluctuation analysis.

## Methods

**A brief summary of Supplementary Information.** Derivations of the CFT, or Eq. (1), are presented in Supplementary Methods, including the derivation of Eqs. (2) and (3) from Eq. (1). The results are summarized in Fig. 1. The methods used in quantitative analyses of the experimental data shown in Figs. 2 and 4 are presented in detail in Supplementary Notes 6, 15, 18, and 21. In Supplementary Note 15, we include the derivation of the analytic results of the mRNA noise for the two transcription models in Fig. 4a, b, used in the quantitative analysis of the experimental data shown in Fig. 4c, d. The stochastic simulation methods used in Figs 3 and 5 are described in Supplementary Note 19. Supplementary Note 4 provides the generalization of the CFT into the case where the product lifetime distribution is strongly heterogeneous among the cells, considered in Fig. 5f.

**Data sources.** The experimental data analyzed in Figs. 2 and 4 are taken from refs. 36 and 23, respectively.

**Code availability.** The Mathematica and C-language code used to generate the reported results are available from the corresponding authors upon request.

**Data availability.** All data used in the current research are available upon request to the corresponding authors.

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

## Acknowledgements

We gratefully acknowledge Professors James J. Collins, Ido Golding, Jaekyung Kim, and Hye Ran Koh for their helpful comments and Mr. Luke Bates for his careful reading of our manuscript. This work was supported by the Creative Research Initiative Project program (2015R1A3A2066497); the NRF grant (MSIP) (2015R1A2A1A15055664); and the Priority Research Center Program through the NRF (2009-0093817), funded by the Korean government.

## Author contributions

J.S. derived the Chemical Fluctuation Theorem. S.J.P., J.-H.K., and J.S. provided the new model and method of analysis; S.J.P., S.S., and J.-H.K. performed analysis; S.S and G.-S. Y. performed stochastic simulations; S.J.P., S.S., P.M.K., S.Y., J.-H.K., and J.S. wrote the

manuscript; S.Y., J.-H.K., and J.S. designed the research. J.S. supervised the entire research project.

## Additional information

**Competing interests:** The authors declare no competing financial interests.

