## [Peer Review File · Nature Communications]

Reviewers' comments:

Reviewer #1 (Remarks to the Author):

The manuscript entitled "Chemical fluctuation theorem governing gene expression." provide an equation (without proof) that relates variability in copy number of molecules for a general birth and death process, in which the production rate can fluctuate. The authors then use this expression to produce analytical results for mRNA noise in few models of transcription. The most complete model includes, promoter switching, variation in the number of genes and fluctuations in the transcription rate. They then analyse three recently published E Coli datasets and provide some novel explanations for the dependence of non-poisson mRNA noise and mean mRNA. The manuscript in its current format has a fundamental problem in its aim. The title promises to be about the "Chemical fluctuation theorem", however, the authors do not provide proof and sufficient insight on the validity and novel aspects of the formula (see below). Instead, they mostly focus on the application of the theorem to existing data. While this is potentially interesting, we have some concerns about the robustness of the conclusions drawn (see below). Overall, this is a potentially interesting work, but in its current format it is too long and not very clear.

Please see below some specific comments:

1 - A derivation needs to be included. Alternatively, the paper about the derivation should be published first. It is hard to judge the novelty of the new Theorem given that the authors do not provide a derivation of this result. The authors note that the theorem "cannot be easily derived from the chemical master equation or its extensions". Such a statement and the enormous amount of references to unpublished material in the supplement render the assumptions of the theorem opaque. Therefore, it is impossible to assess the generality of the result and to which situation the theorem applies.

2- It remains unclear to me what the advantage of the theorem is compared to existing methods? Since the theorem "cannot be easily derived from the chemical master equation or its extensions", it is interesting to note that the authors exclusively make use of models that can be formulated in terms of Master Equations or variants with distributed delays, see for instance Leier, Marquez-Lago. "Delay chemical master equation: direct and closed-form solutions." Proc. R. Soc. A. 471. Interface 2015, or processes described by master equations with time-varying rate constants as in Voliotis, Thomas, Grima, Bowsher (2016). Stochastic simulation of biomolecular networks in dynamic environments. PLoS Comput Biol, 12, e1004923. Moreover, the mRNA transcription and degradation process in which both rate constants are arbitrary stochastic processes, including in the case of static variations, has recently solved exactly for the full distributions and variances: Dattani, Barahona. "Stochastic models of gene transcription with upstream drives: exact solution and sample path characterization." (2017) Interface 14:20160833. I do not see any advantage of using the newly derived theorem over the existing methods for the examples presented.

3- A nice application of the theory is that it allows extracting correlation functions of transcription rates from experimental data. An important conclusion drawn from the first data set (Fig 2) is that the transcription rate must be an oscillatory function with an extremely short period. However, few aspects of this procedure remained unclear to me. How is the function $\Delta(x)$ calculated from the experimental data? How accurately the method represents the data and how well it supports the conclusions? Clearly, there must remain uncertainty in the parameters used.

The authors assume a particular non-monotonic form of $\Delta(x)$ to fit the data. Most of its non-monotonic shape relies on two data points obtained for very small induction levels. It should certainly be possible to reconstruct the correlation function directly from the raw data of $\Delta(x)$, computing their inverse Laplace transform and compare the results of different methods used.

Non-parametric interpolation could also be used avoiding additional assumptions on the data. Comparing the results of both methods should give a more reliable estimate of whether the experimental data supports the conclusions.

In this context, it appears surprising to me that variations in the transcription on the timescale of seconds (Fig. 2) can be inferred from dynamics measured on timescales of minutes (mRNA decay). Intuitively, one would assume that any fast transcription dynamics should be averaged out by downstream processes. Can the authors give provide more explanation on this success?

4- An important source of noise considered in the manuscript is the gene copy number and the induced correlation in the expression levels. Gene copy number is assumed to vary on timescales longer than all other processes, and thus they effectively vary only from cell to cell. Typical timescales of gene copy number variations should be on the order of the cell cycle, however. I wonder how the results of the authors compare to dynamical models of replication as previously presented including analytical results in Peterson, Cole, Fei, Ha, Luthey-Schulten (2015). Effects of DNA replication on mRNA noise. PNAS 112, 15886-15891 and Swain, Elowitz, Siggia (2002). Intrinsic and extrinsic contributions to stochasticity in gene expression. PNAS 99, 12795-12800.

5. In the model in Figure 3, the authors do not consider the possibility of multiple RNAPS transcribing genes at the same time. However, this is a likely possibility, highlighted by the common TASEP models of transcription (and translation). How sensitive are the results to this assumption.

6. The authors assume, all genes have the same gene expression parameters in figure 4, which sounds a very unrealistic assumption. Could they motivate this assumption better.

6. The work seems highly related to reference 46. The connection and novelty of the results should be better discussed.

Reviewer #2 (Remarks to the Author):

Overview: This paper presents the Chemical Fluctuation Theorem (CFT), a theorem stating the relationship between cellular environment and gene expression variability. The authors derive the predictions that this theorem makes about mRNA variability and its associated noise sources, given set of transcriptional models. They then check how these predictions fit data from three existing experimental datasets, which are:

1. *E. coli* cell-to-cell variability in number of mRNA expressed from *lacZ*, for various IPTG concentrations (Reference 41).
2. mRNA and protein number statistics for a comprehensive set of *E. coli* genes (Reference 42).
3. mRNA copy number distribution of a library of synthetic promoters driving *lacZ*, in *E. coli* (Reference 26). Here the differences in promoter sequences have clear interpretations in terms of the molecular parameters underlying transcription (e.g., transcription factor unbinding rate, basal transcription rate). This allows the researches to link the molecular events underlying transcription with the observed variability in gene expression.

Recommendation : The paper hinges completely on one equation (i.e. equation (1) in the paper) which states the CFT. Even though this equation is explained in detail, without a proper mathematical proof it is hard to ascertain the conditions under which this equation holds and examine the scientific claims made by the paper. For example, it is difficult to see if the equation will remain valid if there is feedback from mRNA to the gene-switching rates. The analysis of existing experimental data based on CFT reveals interesting conclusions on the sources contributing to mRNA variability. However, it is not checked whether the conclusions hold for data that has not been used to fit the models. For instance, it is mentioned that RNAP binding activity fluctuates at a rate ≥ 100 Hz for constitutive promoters. This conclusion is extracted from analyzing the data in Reference 26, but without any other cross-checks, it is hard to assess the generality of this claim.

Overall, the paper is hard to assess in terms of its scientific contributions. The central result CFT is not presented with enough detail in this work. The paper would gain considerably if more mathematical details about how CFT is derived and its underlying assumptions is provided. The other important issues that need to be addressed are mentioned below.

1 Major Issues

1. Gene-expression variability comes from both transcription and translation steps. The paper focuses extensively on transcriptional variability but how this variability is filtered through the translation step is not properly analyzed.
2. It seems that the main advantage of CFT is that it produces a *noise factorization* of mRNA variability (like equation (2) in the paper), from which Poisson and non-Poisson part of the variability can be evaluated and the non-Poisson part can be further analyzed in terms of its contributing factors. However such factorizations are likely to fail if mRNAs, or its proteins, somehow influence the transcription rate $\kappa(\Gamma)$ or the gene-activity fluctuations ξ (see lines 231-234 in the text). This appears to be a major limitation of the usefulness of the analysis presented in this paper as many gene-expression networks consist of transcriptional feedbacks. Please discuss this issue in the paper in detail.
3. The paper cites two key papers (references 33 and 63) that quantitatively analyze the contributions to noise from intrinsic and extrinsic sources. Are these noise decomposition results related to the noise factorizations present in this paper? This issue must be investigated in detail.
4. While checking the CFT-predictions with the three existing experimental datasets, the authors estimate certain parameters of their model by fitting them to the data and use values from the literature for the other parameters. Here it is important for the authors to provide some comments on the model-fitting process and the accuracy of the estimated parameters. It would additionally be very interesting to see how the resulting model fits data it has previously not seen. These comments are needed to assess the generality of the conclusions exposed (whether the match between experimental data and theoretical CFT-based predictions is not simply due to model *overfitting*).

2 Minor comments

1. Please explain terms like *non-Poisson transcription*, *sub-Poisson transition* in the introduction.

Reviewer #3 (Remarks to the Author):

In its current form, the article under review presents results that are derived from Equation (1). This equation is presented without proof, but instead proclaimed with a promise that "a rigorous derivation of equation will be presented elsewhere shortly."

For publication in Nature journals multiple criteria need to be met, but the most basic one is that a manuscript must present "strong evidence for its conclusion" as explicitly stated by the publisher. Without presenting a proof of Equation (1) the conclusions presented in this manuscript cannot be deemed to have sufficient evidence. The paper in its current form can thus only be rejected.

Response to **Reviewer 1**

We are grateful to Reviewer 1 for his comments that have greatly helped us improve our manuscript. The following are our responses to each comment by Reviewer 1:

1. A derivation needs to be included. Alternatively, the paper about the derivation should be published first. It is hard to judge the novelty of the new Theorem given that the authors do not provide a derivation of this result. The authors note that the theorem “cannot be easily derived from the chemical master equation or its extensions”. Such a statement and the enormous amount of references to unpublished material in the supplement render the assumptions of the theorem opaque. Therefore, it is impossible to assess the generality of the result and to which situation the theorem applies.

Response: As Reviewer 1 requested, we present the mathematical derivation of the chemical fluctuation theorem (CFT) in Supplementary Method 1. The derivation of the CFT presented in Supplementary Method 1 requires no assumption regarding the stochastic properties of the product creation process and the lifetime distribution of product molecules. This means that the CFT holds even when the product creation process is subject to a feedback regulation or other regulation mechanisms and when the product decay process is an arbitrary non-Poisson process. The only assumption we make in the derivation is that the lifetimes of product molecules are identically distributed, independent random variables. It is possible to extend the CFT to encompass a more general situation, but CFT would then lose its concise form.

2. *It remains unclear to me what the advantage of the theorem is compared to existing methods? Since the theorem “cannot be easily derived from the chemical master equation or its extensions”, it is interesting to note that the authors exclusively make use of models that can be formulated in terms of Master Equations or variants with distributed delays, see for instance Leier, Marquez-Lago. “Delay chemical master equation: direct and closed-form solutions.” Proc. R. Soc. A. 471. Interface 2015, or processes described by master equations with time-varying rate constants as in Voliotis, Thomas, Grima, Bowsher (2016). Stochastic simulation of biomolecular networks in dynamic environments. PLoS Comput Biol, 12, e1004923. Moreover, the mRNA transcription and degradation process in which both rate constants are arbitrary stochastic processes, including in the case of static variations, has recently solved exactly for the full distributions and variances: Dattani, Barahona. “Stochastic models of gene transcription with upstream drives: exact solution and sample path characterization.” (2017) Interface 14:20160833. I do not see any advantage of using the newly derived theorem over the existing methods for the examples presented.*

Our response: We thank the reviewer for this insightful comment. In response to this comment, we now discuss these methods in detail in the revised manuscript; we mention these papers as references 62, 63, and 64, and additionally present a brief review of each of them in Supplementary Note 4.

In particular, our Chemical Fluctuation Theorem (CFT) is applicable to intracellular network models more general than the transcription network models used in these works. For example, it is applicable to the cases where the product creation process is under various types of regulation processes and the lifetime distribution of product molecule is an arbitrary non-exponential distribution. To the best of our knowledge, none of the previously reported theories yields exact results for the more general cases that are frequently encountered in reaction

networks in living cells. In this work, we use models where the mRNA lifetime distribution is a simple exponential, not because the range of our theory is limited to the case where the product lifetime distribution is an exponential distribution, but because the experimental data presented in reference 44 show that the mRNA lifetime distribution can be well approximated within experimental errors by such a distribution. Although, in this paper, we apply CFT only to the transcription systems where the mRNA lifetime distribution is exponential, we believe that the greater application range of our result is more of a merit than a drawback; our result provides exact results for reaction networks producing product molecules with non-exponential lifetime distributions, which cannot be accurately described by the master equation or its variants.

In the delay chemical master equation (DCME) introduced by Leier and Marquez-Lago (reference 62 in the revised manuscript), a complex chemical process is modeled as a single-delay reaction. Here, the single delay reaction is characterized by the distribution of time delays or elapsed times taken to complete a product creation event after the reaction is initiated, so that DCME is more general than the conventional chemical master equation in the sense that the single delay reaction can represent a non-Poisson reaction process. However, the derivation of a solvable DCME for general networks including feedback loops remains a difficult task, though the stochastic simulation of the single delay reaction is possible. Voliotis, Thomas, Grima, and Bowsher suggested different simulation algorithm for reaction networks in dynamically fluctuating environments (reference 63 in the revised manuscript). This method simulates trajectories that can be obtained from the chemical master equation with time-dependent, stochastic rates. However, the authors' simulation method is not directly applicable to regulatory networks containing feedback regulation. Neither of these approaches gives the general analytical result for second-order chemical

fluctuation, which is the central result of our work.

Extending Gardiner and Chaturvedi's Poisson mixture ansatz into the case where the mean of a Poisson distribution is governed by the first-order differential equation with time-varying, stochastic creation and degradation rates, Dattani and Barahona obtained the general relationship between the product (mRNA) number moments and Poisson mean moments (reference 64 in the revised manuscript). Among them, the second-order moment equation is comparable to our CFT, but the application range of their equation is essentially limited to the case where the degradation rate is a constant or a deterministic function of time. When the rate of the product degradation process is constant, the second-order moment equation reduces to the result in reference 53 in the steady state. One can consider a more general case with a stochastic degradation rate, for which an explicit analytic result is missing in reference 64. Taking the approach in ref. 64, one can derive a formal expression of the second moment of the product number (Supplementary reference 36 in the revised manuscript). However, to obtain an explicit analytic result from the formal expression, one must have the analytic expressions for the multi-time correlations between stochastic rates governing the time evolution of the Poisson means up to the infinite order, which makes the practical application of this approach infeasible when the product lifetime distribution is an arbitrary non-exponential function. To the best of our knowledge, the CTF, equation (1), reported in this work cannot be obtained by taking the previously reported approaches.

It is remarkable that, as demonstrated in reference 64, the analytic result of *the time-dependent mRNA number distributions* can be obtained for simple models by solving the time-evolution equation of the distribution of the Poisson mean. This equation conforms to the generalized Fokker-Planck equation describing general vibrant reaction networks considered in ref. 53. These equations, however, are not applicable to reaction networks with a feedback regulation. Even for the gene

expression network without any feedback regulation, it is not feasible to provide a quantitative explanation of the experimental data for gene expression statistics with use of the models considered in ref. 64.

As far as we are aware, a unified, quantitative understanding of the experimental results obtained for various transcription systems, which we achieve in the present work, is unprecedented. This was made possible because we have used a new type of transcription model, which involves implicit but complete modeling for the environmental variable dependent rate factors, and because we take into account gene-copy number variation, the correlation between the mRNA levels transcribed from different gene copies, and the non-Poisson transcription dynamics of each gene copy. This model has not been considered in any of the references mentioned in this comment. We believe our model and the new analysis method introduced in this work are novel and remarkably useful.

3A. *A nice application of the theory is that it allows extracting correlation functions of transcription rates from experimental data. An important conclusion drawn from the first data set (Fig 2) is that the transcription rate must be an oscillatory function with an extremely short period. However, few aspects of this procedure remained unclear to me. How is the function $\Delta(x)$ calculated from the experimental data? How accurately the method represents the data and how well it supports the conclusions? Clearly, there must remain uncertainty in the parameters used. The authors assume a particular non-monotonic form of $\Delta(x)$ to fit the data. Most of its non-monotonic shape relies on two data points obtained for very small induction levels. It should certainly be possible to reconstruct the correlation function directly from the raw data of $\Delta(x)$, computing their inverse Laplace transform and compare the results of different methods used. Non-parametric interpolation could also be used avoiding additional assumptions on the data. Comparing the results of both methods should give a more reliable estimate of whether the experimental data supports the conclusions.*

Our response: In response to the Reviewer 1's comment, we present a clearer explanation of the fitting procedure for $\Delta(x)$ in Supplementary Methods 3 (previously 2). We have performed new analysis to assess the standard errors of the extracted parameter values and present the result in Table S1-S3. As Reviewer 1 suggested, we have also carefully repeated our analysis with the use of the non-parametric interpolation of the raw data version of $\Delta(x)$ and we present the results in Supplementary Figure S2.

In the newly added analysis, we have used the Durbin-Crump method for the numerical inverse-Laplace transform in extracting the time correlation function (TCF) of the active gene transcription rate from the non-parametric interpolation of the raw data version of $\Delta(x)$. The resulting TCF shows an oscillatory feature in

qualitative agreement with the result of our analysis that relies on the smooth function version of $\Delta(x)$ representing a global trend in the data.

We have also used the Stehfest method for the numerical inverse Laplace transform to extract the TCF of the transcription rate from the non-parametric interpolation of the raw data version of $\Delta(x)$. In contrast with the TCF obtained from the Durbin-Crump method, the TCF extracted from the Stehfest method has a noisy shape and the details of the shape depend on which options are chosen for the numerical inverse Laplace transform routine in use. However, we find that the noisy TCF extracted from the Stehfest method also show the oscillatory feature in qualitative agreement with the TCF extracted with use of the Durbin-Crump method, or the result of our analysis that relies on equation (M3-13), the smooth function version of $\Delta(x)$, which are presented in Figure S2 and Figure 2, respectively. However, we do not present the unnaturally irregular TCFs extracted using the Stehfest method in the figure because it does not reflect an inherent property of the transcriptional system. Given that the variance in the copy number of mRNA is a slowly varying function of the mean mRNA number, $\Delta(x)$ and hence the TCF of the transcription rate should be smooth functions.

3B. *In this context, it appears surprising to me that variations in the transcription on the timescale of seconds (Fig. 2) can be inferred from dynamics measured on timescales of minutes (mRNA decay). Intuitively, one would assume that any fast transcription dynamics should be averaged out by downstream processes. Can the authors give provide more explanation on this success?*

Our response: As Reviewer 1 noted in this comment, in the limit of the fast transcription rate fluctuation, the mRNA noise approaches the Poisson noise independent of the transcription dynamics. Figures 2a in the main manuscript show a significant deviation of the steady-state mRNA noise from the Poisson noise. These experimental data indicate that, for the transcription systems investigated in the present work, the effects of the non-Poisson transcription dynamics are not negligible and cause mRNA noise to deviate from the Poisson noise to the extent that it can be quantitatively measured.

The successful quantitative explanation of the stochastic transcription in terms of transcription dynamics could be achieved, in part, by focusing on the analysis of the non-Poisson mRNA noise component rather than the analysis of the entire mRNA noise. This is because the non-Poisson noise component is far more sensitive to the transcription dynamics. If the fluctuation in the active gene transcription rate, the TCF of which is shown in Figure 2b, were the only source of the non-Poisson mRNA noise, the value of non-Poisson mRNA noise would have been far smaller than the experimentally measured values. However, the non-Poisson mRNA noise is also contributed from the transcription rate fluctuation caused by the gene-state switching process. As shown in Supplementary Figure S14b, the mRNA noise, $\chi_{n\xi}\eta_\xi^2$, originating from the gene-state switching process and the mRNA noise, $\chi_{n,(\kappa,\xi)}\eta_\kappa^2\eta_\xi^2$, from the bilinear coupling term make far greater

contributions to the non-Poisson mRNA noise than the mRNA noise, $\chi_{nk}\eta_k^2$, from the fluctuation in the active gene transcription rate.

In response to this comment by Reviewer 1, we have inserted Supplementary Figure 16 to make this content more accessible to readers. The transcription rate fluctuation due to the gene-state switching between the active and inactive gene states has a different stochastic property from the rate fluctuation of the active-gene transcription that is a multi-step, consecutive reaction process. In general, the product noise of a multi-channel reaction, such as transcription with the gene-state switching process, is greater than the product noise of a single-channel reaction, even if one of the channels is inactive as is the case here. On the other hand, the product noise of a multi-step reaction is smaller than the product noise of a single step reaction, because the randomness in the time taken to complete a reaction process decreases as the number of the intermediate reaction steps composing the reaction increases.

In terms of mathematics, the transcription rate fluctuation originating from the gene-state switching shows a monotonically decaying TCF, $\phi_\xi(t)$, but the active gene transcription rate fluctuation shows an oscillating TCF, $\phi_k(t)$. According to CFT or equation (2), the mRNA noise, $\chi_{n\xi}\eta_\xi^2$, originating from the gene-state switching process is related to its TCF by $\chi_{n\xi}\eta_\xi^2 = \gamma \int_0^\infty dt e^{-\gamma t} \phi_\xi(t) \eta_\xi^2$. Similarly, the mRNA noise, $\chi_{nk}\eta_k^2$, originating from active gene transcription rate fluctuation is given by $\chi_{nk}\eta_k^2 = \gamma \int_0^\infty e^{-\gamma t} \phi_k(t) \eta_k^2$. As shown in Supplementary Figure S1b, $\chi_{nk}\eta_k^2$ including the integration of the oscillating TCF, $\phi_k(t)$, is far smaller than $\chi_{n\xi}\eta_\xi^2$, including the integration of the monotonically decaying TCF, $\phi_\xi(t)$. This can be the case even when η_k^2 is far greater than η_ξ^2 . The non-Poisson mRNA noise contributed from the bilinear coupling term is given by

$\chi_{n,(\kappa,\xi)} \eta_\kappa^2 \eta_\xi^2 = \gamma \int_0^\infty dt e^{-\gamma t} \phi_\kappa(t) \phi_\xi(t) \eta_\kappa^2 \eta_\xi^2$, which can be either super-Poisson or sub-Poisson noise depending on the shape of $\phi_\xi(t) \phi_\kappa(t) \eta_\kappa^2 \eta_\xi^2$.

4. An important source of noise considered in the manuscript is the gene copy number and the induced correlation in the expression levels. Gene copy number is assumed to vary on timescales longer than all other processes, and thus they effectively vary only from cell to cell. Typical timescales of gene copy number variations should be on the order of the cell cycle, however. I wonder how the results of the authors compare to dynamical models of replication as previously presented including analytical results in Peterson, Cole, Fei, Ha, Luthey-Schulten (2015). Effects of DNA replication on mRNA noise. PNAS 112, 15886-15891 and Swain, Elowitz, Siggia (2002). Intrinsic and extrinsic contributions to stochasticity in gene expression. PNAS 99, 12795-12800.

Our response: We are grateful to Reviewer 1 for informing us of these interesting papers. The non-Poisson mRNA noise, equation (3), accounting for the effect of gene copy number variation is obtained by combining equations (M2-7a) and (M2-7b). For convenience, both equations are reproduced below:

$$\langle n \rangle = \langle g \rangle \langle n \rangle_1 \quad (\text{M2-7a})$$

$$\langle n^2 \rangle = \langle g^2 \rangle \langle n \rangle_1^2 + \langle g \rangle \sigma_{n,1}^2 + \langle g(g-1) \rangle c \quad (\text{M2-7b})$$

These equations are valid irrespective of the explicit time dependence of the slow gene copy number variation. When the gene copy number, g , is either 1 or 2, Jones *et al.* in reference 29 obtained the following equations for $\langle g \rangle$ and $\langle g^2 \rangle$:

$$\langle g \rangle = 1 + f \quad (\text{R1})$$

$$\langle g^2 \rangle = 1 + 3f \quad (\text{R2})$$

where f denotes the fraction of cell cycle after gene duplication. Later, Peterson *et al.* (now reference 67) found the extended version for equations (R1) and (R2) with dynamic correction accounting for the effect of mRNA degradation:

$$\langle g \rangle = 1 + f + \frac{e^{-f\gamma\tau} - 1}{\gamma\tau} \quad (\text{R3})$$

$$\langle g^2 \rangle = 1 + 3f + \frac{8e^{-f\gamma\tau} - 2e^{-2f\gamma\tau} - 7}{2\gamma\tau} \quad (\text{R4})$$

where γ and τ denote the inverse lifetime of mRNA and cell doubling time, respectively. The dynamic correction explicitly indicates the third term on the right-hand side of either equation (R3) or equation (R4). In the large $\gamma\tau$ limit, equations (R3) and (R4) reduce to equations (R1) and (R2). In other words, equations (R1) and (R2) are valid for large $\gamma\tau$, which is usually the case. Before this, Swain *et al.* (reference 66) also developed a time-dependent theory but the authors estimated the dynamic correction to be negligible, which can be attributed to the fact that the values of the relevant parameters they used fall into the case of large $\gamma\tau$.

Although we used the time-independent theory in reference 29 in the calculation of the mean and variance of the gene copy number, this issue does not pose a problem because the value of $\gamma\tau$ is suitably large. For example, the value of $\gamma\tau$ is estimated to be 30 with $\gamma^{-1} = 2$ min and $\tau = 60$ min for the constitutive expression data in Fig. 5. In this case, the relative deviations of equations (R1) and (R2) from equations (R3) and (R4) are estimated to be 2% and 4%, respectively. For the inducer-controlled expression data we used in Fig. 2, where g is either 2 or 4, none of equations (R1)-(R4) is not directly available, because these equations are derived for the case where g is either 1 or 2. However, we could still estimate the values of $\langle g \rangle$ and $\langle g^2 \rangle$ for the experimental data in Fig. 2 as shown in Supplementary Note 13.

In response to the Reviewer 1's comment, we have added Supplementary Note 7 detailing the above content.

5. *In the model in Figure 3, the authors do not consider the possibility of multiple RNAPS transcribing genes at the same time. However, this is a likely possibility, highlighted by the common TASEP models of transcription (and translation). How sensitive are the results to this assumption.*

Our response: In response to the Reviewer 1's comment, we have revised the caption of Figure 3 to present a more detailed account of our transcription model.

Our transcription model takes into account the situation where multiple RNAPs transcribe a single gene at the same time. This is described in Supplementary Method 7 in the revised manuscript, where we present the simulation algorithm of our transcription model represented by the scheme in Fig. 3a. At the transcriptional initiation step of the RNAP-promoter complex, the next round of RNAP binding to the promoter is not allowed before the preceding RNAP leaves the promoter DNA to complete successful initiation. During the transcription elongation by a RNAP, other RNAP can associate with the promoter and proceed to the next step. In the steady state, a battery of RNAPs simultaneously undergoes transcriptional elongation along each gene copy.

6. The authors assume, all genes have the same gene expression parameters in figure 4, which sounds a very unrealistic assumption. Could they motivate this assumption better.

Our response: As noted by Reviewer 1, gene expression parameters differ from gene to gene. However, the genome-wide data shown in Figure 4 clearly show the global trend in the dependence of the mRNA noise on the mean mRNA. Gene-specific deviations from the global trend also exist because of gene-to-gene variation in the regulation mechanism and gene-expression parameters. However, the gene-specific deviations are not so large that the global trend in the transcription statistics is easily noticeable.

The purpose of our analysis in Fig. 4 is not to show that we can explain the entire genome-wide data for mRNA noise vs. mean mRNA using the same gene expression parameters. As mentioned in the last paragraph in Analysis 2 on page 19, our analysis in Fig. 4 shows that the k_{off} modulation mechanism is not the universal transcription-control mechanism of *E. coli* genes as suspected in refs. 44 and 72.

Quantitative analysis of a global trend in gene expression statistics with use of a single gene expression model has been of interest in the previous literature, for example, in references 45, 53, and 72.

7. *The work seems highly related to reference 46. The connection and novelty of the results should be better discussed.*

Our response: In response to comment 7, we present a more detailed discussion about the novelty of the present work and its relationship to reference 53 (previously 46) in Supplementary Note 3.

- The chemical fluctuation theorem (CFT), equation (1), in the present work has a greater application range over the key result in ref. 53, which can be summarized as follows:

1) CFT in the present work is applicable to biological networks with an arbitrary regulation mechanism on the product creation process, whereas the result in ref. 53 is not. The result in ref. 53 is only applicable to those biological networks in which the product creation process is not dependent on the product number.

2) CFT in the present work is applicable to both a non-stationary product creation process as well as a stationary process, whereas the result in ref. 53 is only applicable to the latter.

3) CFT in the present work is applicable to the case where the lifetime distribution is a non-exponential function, to which the result in ref. 53 cannot be applied.

- In the present work, we apply CFT to the quantitative analysis of mRNA variability among a clonal population of cells for three different experimental data, namely those published in refs. 29, 44, and 45. In contrast, the authors of ref. 53 mainly focused on the application of their result to the quantitative analysis of protein level variability in the dual reporter system, reported in ref.

52. Since researchers in each experiment employed a different control variable, we use different models accordingly in the present work. These models are also different from the model used in ref. 53.

- In the present analysis, the effect of gene copy number variability is explicitly modelled with the use of information extracted from experimental data reported in refs. 29, 44, and 45. By doing so, we take great strides in achieving a separate estimation of mRNA noise originating from gene copy number variation and various other sources. In contrast, in ref. 53, the effects of gene copy number variability were implicitly taken into account with gene copy number treated as a hidden variable.

Response to Reviewer 2

We are grateful to Reviewer 2 for his or her criticisms and questions that have helped us improve our manuscript as well as for assessing our work as interesting.

The following is our response to each comment made by Reviewer #2:

Recommendation

1. *For example, it is difficult to see if the equation will remain valid if there is feedback from mRNA to the gene-switching rates.*

Our response: In response to the Reviewer 2's comment, we present the detailed derivation of the chemical fluctuation theorem (CFT) in Supplementary Method 1 in the revised manuscript. As shown in the derivation, the correctness of CFT does not rely on particular properties of the product creation process. This means that CFT holds regardless of any regulation mechanism on the product creation process.

2. *For instance, it is mentioned that RNAP binding activity fluctuates at a rate ≥ 100 Hz for constitutive promoters. This conclusion is extracted from analyzing the data in Reference 29, but without any other cross-checks, it is hard to assess the generality of this claim.*

Our response: We are grateful for this comment by Reviewer 2. We have referenced other publications that further support the result of our analysis in the 8th line from the bottom of page 6 in the main text and in Supplementary Note 1:

Briefly, it was previously shown that the supercoiling state of DNA is coupled to the formation of the pre-initiation complex and subsequent initiation process [Gilbert and Allan, "Supercoiling in DNA and chromatin", *Current Opinion in Genetics & Development* **25**, 15 (2014) and Corless and Gilbert, "Effects of supercoiling on chromatin architecture", *Biophysical reviews* **8**, 245 (2016)]. Such a tendency differs from gene to gene or depends on the promoter sequence [Wagner, "Transcription regulation in prokaryotes", (Oxford University Press, New York, 2000) and Wang, "DNA supercoiling and gene expression" in Pullman, Ts'O, and

Schneider, "Interrelationship among aging, cancer, and differentiation", (Springer, Netherlands, 1985)]. The time scale associated to non-enzymatic supercoil dynamics amounts to 10 ms or less, which is consistent with our estimation of the RNAP-promoter binding affinity fluctuation time scale [Crut et al., "Fast dynamics of supercoiled DNA revealed by single-molecule experiments", PNAS **104**, 11957 (2007) and Koster et al., "Cellular strategies for regulating DNA supercoiling: a single-molecule perspectives", Cell **142**, 519 (2010)].

3. The paper would gain considerably if more mathematical details about how CFT is derived and its underlying assumptions is provided.

Our response: As Reviewer #2 suggested, we present a detailed mathematical derivation of CFT in Supplementary Method 1. As shown in the derivation, CFT is a general result that can be derived without any assumption about the property of the product creation process. This means that it holds exactly for any intracellular regulatory network in which the product creation rate is modulated by product number or any other environmental variables. The only assumptions involved in our derivation of CFT is that the lifetimes of product molecules are identically distributed, independent random variables and the product lifetime distribution does not change over time. It is possible to think of a more general product decay process, but CFT in this case would not have a concise form and would become far more complicated. We believe that the current form of CFT is already general enough to provide a quantitative explanation of the chemical fluctuation in most intracellular networks.

Major Issues

1. *Gene-expression variability comes from both transcription and translation steps. The paper focuses extensively on transcriptional variability but how this variability is filtered through the translation step is not properly analyzed.*

Our response: As Reviewer #2 correctly noted, in the present work, we apply the CFT to quantitative analyses of the transcription level variability among a clonal population of cells for three different experimental systems. This is partly due to the fact that protein level variability data were missing for two of the three experimental systems. However, we agree that it is important to discuss how the mRNA level variability propagates into the protein level variability. Addressing this issue, we have added Supplementary Note 2. As discussed in Supplementary Note 2, CFT is applicable to the translation process as well. By applying CFT to the translation process, one can achieve a general understanding of how the variation in the mRNA level propagates into the variation in the protein level.

We believe that a full quantitative analysis of translation would be out of the scope of this current paper. Instead, we have referred to reference 53 (previously 46), where the authors applied an earlier, less general version of CFT to the translation process as well as to the transcription process in order to quantitatively analyze the dependence of the mean and variability of protein levels on the mean and variation in the RNAP level.

2. However such factorizations are likely to fail if mRNAs, or its proteins, somehow influence the transcription rate $\kappa(\Gamma)$ or the gene-activity fluctuations ξ . This appears to be a major limitation of the usefulness of the analysis presented in this paper as many gene-expression networks consist of transcriptional feedbacks. Please discuss this issue in the paper in detail.

Our response: Reviewer #2's concern raised in this comment is actually relevant to the fluctuation theorem derived in reference 53 (previously 46). We believe that the derivation appended in Supplementary Method 1 in the revised manuscript will also resolve the issue raised in this comment. As clearly shown in the derivation, now included in our revised manuscript, our CFT holds exactly, regardless of the stochastic properties of and the regulation mechanism on the product creation process. This means that our result is correct, even if mRNAs, or their proteins, somehow influence the transcription rate, $\kappa(\Gamma)$, or the gene-activity fluctuations, ξ .

In response to the Reviewer #2's comment, in Supplementary Note 10, we present a brief account explaining the validity of CFT for the transcription process under a feedback regulation. The mathematical structure of the CFT, given in equation (1), remains the same in the presence of a feedback regulation, independent of the detailed nature of the regulation mechanism. However, the transcription rate, R , which appears in CFT, can be dependent on the number of mRNAs or proteins, i.e.,

$$R = R_{TX} = k_{TX} \theta(m, p)$$

where $\theta(m, p)$ is the transcription rate factor with a mathematical form dependent on the details of the regulation mechanism. For example, for a feedback transcription network, rate factor θ can take the Hill-type form:

$$\theta = \frac{K(p)N_{Rp}}{1 + K(p)N_{Rp}}$$

where the RNAP-promoter binding affinity, $K(p)$, is dependent on the protein copy number, p , given by

$$K(p) = \frac{K_0}{1 + K_p p^h}$$

Here, K_0 , K_p , and h denote the RNAP-promoter binding affinity in the small p limit, the binding affinity of protein to the operator site, and the Hill exponent, respectively. Due to coupling to the cell-state variables, K_0 and K_p are not just simple constants but stochastic variables. A positive h would then indicate negative feedback, and a negative h , positive feedback.

In the actual application of the CFT to the quantitative analysis of the chemical fluctuation resulting from a regulatory network, it is necessary to calculate the time-correlation function of the product creation rate, which depends on the product number. To calculate the time-correlation function of the transcription rate under a feedback regulation, one can use various levels of mathematical methods. Finding the optimal mathematical model is a goal we leave to future research.

3. *The paper cites two key papers (references 33 and 63) that quantitatively analyze the contributions to noise from intrinsic and extrinsic sources. Are these noise decomposition results related to the noise factorizations present in this paper? This issue must be investigated in detail.*

Our response: The chemical noise in living cells is often written as the sum of two components, intrinsic noise and extrinsic noise, by researchers in this field. However, in the literature, there has been controversy regarding the most appropriate definition of intrinsic noise and extrinsic noise. This issue was thoroughly examined in references 37 and 53 (previously 46).

According to the CFT, equation (1), the noise in the product number, or the product noise, can be separated into a Poisson noise component, the first term on the right-hand side of equation (1), and a non-Poisson noise component, the second term. Here, regardless of the details in the product creation network and its coupling to the cell environment, the Poisson noise component is always given by the inverse of the mean without fail, while, on the other hand, the dependence of the non-Poisson noise component on the mean *is* dependent on these details. The Poisson noise component can be thought of as universal “intrinsic noise”. The non-Poisson noise component can then be thought of as “extrinsic noise” and any remaining, non-universal or system-dependent “intrinsic noise”. However, in CFT, equation (1), there is no distinction between extrinsic noise and the non-universal, system-dependent intrinsic noise, so that both are taken into account as a single term in a unified manner. A further separation of the non-Poisson noise component between extrinsic noise and non-universal, system-dependent intrinsic noise depends on one’s definition of intrinsic noise and extrinsic noise.

In the present work, instead of separating product noise into intrinsic and extrinsic noise, we have factored the single gene transcription rate into two factors: the control variable dependent factor and the environmental variable dependent factor. The former takes into account the rate of the chemical process that is coupled to the experimentally controlled variable and possibly to the environmental variable as well. On the other hand, the latter takes into account the

rate of the remaining chemical processes in the network, which are coupled to the environmental variables, but not to the control variable. Using the factorized form of the product creation rate in CFT, we can obtain equation (2) for the relationship between the mRNA noise and the noise in both rate factors of the single gene transcription version of Model III. By further considering the multi gene transcription version of Model III, we obtain equation (3). As shown in equations (2) and (3), fluctuations in both the control variable and the environmental variable dependent rate factors contribute to the non-Poisson component of the product noise. However, the product noise is not given by the sum of the noise arising from fluctuation in the two rate factors, but is instead given by a bilinear function. This means that the product noise originating from each of the two rate factors does not represent either intrinsic or extrinsic noise, because the sum of the intrinsic and extrinsic noise yields the total product noise always.

In response to the Reviewer #2's comment, we have added Supplementary Note 5 detailing the above content.

4A. *While checking the CFT-predictions with the three existing experimental datasets, the authors estimate certain parameters of their model by fitting them to the data and use values from the literature for the other parameters. Here it is important for the authors to provide some comments on the model-fitting process and the accuracy of the estimated parameters.*

Our response: As Reviewer #2 suggested, we have provided a more detailed account of the model fitting process in Supplementary Methods 3-5 in the revised manuscript, and have noted the standard error to each of the extracted values of the parameters. The results are summarized in Supplementary Tables 1-3, where we have added more explanations about how the parameters are extracted or enumerated.

4B. *It would additionally be very interesting to see how the resulting model fits data it has previously not seen.*

Our response: We agree. It would be interesting to test and optimize our model of the transcription process against future experimental data.

The general strategy in the application of the CFT to the quantitative analysis of biological noise is to identify the control variable and to construct an accurate, explicit model for the control variable dependent part of the product creation rate, while treating the uncontrollable or hidden environmental variable dependent part of the product creation rate as a general stochastic variable, as we demonstrate in the present work.

Minor comments

1. *Please explain terms like non-Poisson transcription, sub-Poisson transition in the introduction.*

Our response: According to the reviewer's comment, we have added an explanation of the non-Poisson transcription in the following sentence starting at the 7th line from the bottom of page 5 in the revised manuscript:

'This means that the transcription of an activated gene is not a Poisson process with a constant rate, but a non-Poisson process with a stochastic rate; in other words, the distribution of time between successive transcription events is not an exponential function.'

We have replaced "sub-Poisson transition" with "multi-step transition process", a more accessible term to general readers, in the revised manuscript.

Response to Reviewer #3

1. *In its current form, the article under review presents results that are derived from Equation (1). This equation is presented without proof, but instead proclaimed with a promise that "a rigorous derivation of equation will be presented elsewhere shortly." For publication in Nature journals, multiple criteria need to be met, but the most basic one is that a manuscript must present "strong evidence for its conclusion" as explicitly stated by the publisher. Without presenting a proof of Equation (1) the conclusions presented in this manuscript cannot be deemed to have sufficient evidence. The paper in its current form can thus only be rejected.*

Our response: In response to the Reviewer #3's comment, we have appended the mathematical derivation of the Chemical Fluctuation Theorem in Supplementary Method 1.

Reviewers' comments:

Reviewer #1 (Remarks to the Author):

The revised manuscript as addressed several of my original concerns. As explained below some specific points remain unclear. However, at a general level, I am not sure if the manuscript is suitable for Nature Communications. The manuscript presents Chemical Fluctuation Theorem that is applicable to a broad class of birth-death processes (but not other forms of chemical reactions, such as binding). Then the authors apply their new theory to some datasets to provide new insight. However, the models they use can be also handled with existing methods, so the examples fail to show case the strength of the CFT (And why it is very different from other existing methods). The manuscript is also not easily accessible to a wider audience and is extremely long (150 pages of supplements). I think it maybe better to publish the CFT and relevant non-trivial examples in a separate paper and then write a separate paper on the biological examples.

I have the following specific comments in response to the rebuttals.

i) In their response, the authors mention that the CFT applies more generally to feedback regulation, which is an advantage over other methods in the literature. The authors, however, do not demonstrate such an application, and the advance seems to be irrelevant to explain the experimental data. Further, we foresee several issues with the CFT in the case of feedback regulation. Specifically, for feedback regulation, the transcription rate is a potentially nonlinear function of the product number or even of its history. In this case, however, the CFT cannot provide a closed-form characterization of the noise because the correlation terms on the RHS involve higher order moments or correlation functions. Thus the CFT does not lead to a closed system of equation and therefore has similar limitations compared to other methods, e.g. Dattani, Barahona. I cannot see that the simulation methods mentioned by the authors should suffer from the same issue, however.

(ii) The authors also mention that the CFT applies more generally to non-exponential product lifetimes. In their derivation, they assume independent lifetimes. Non-exponential lifetimes should be observed when mRNAs compete for degradation or dependent on common factors. Non-exponential lifetimes should, therefore, be generally correlated. However, as the authors emphasise, non-exponential lifetimes are not observed experimentally! Thus it the advantage of the CFT over other methods seems irrelevant to explain the experimental data.

(iii) The authors point out that transcription rate fluctuations represent a small if not negligible contribution to the noise. This is in agreement with the intuition that fast oscillations in the production rate are integrated by the dynamics. The coupling term represents the biggest contribution to the noise. However, the expression for the coupling terms contains an oscillatory integrand and therefore the biggest contribution seems to stem from the integrand at zero, as shown in Fig S16. We are not convinced that this provides sufficient evidence for the existence of oscillations in the transcription rate. We wonder whether the observed oscillations could not as well be reproduced by white noise correlations of similar amplitude, at least this is what Fig S16 seems to suggest. Perhaps the fast oscillatory behaviour of the transcription rate correlation is an artefact from restricting $\Delta(x)$ to have finite support before computing the inverse Laplace transform? In summary, it has not become clear which feature of $\Delta(x)$ leads the authors to conclude on the presence of high-frequency oscillations and how this feature is represented in the data (Fig S1).

Reviewer #4 (Remarks to the Author):

Overview: The main object of the paper is to present the *Chemical Fluctuation Theorem* (CFT) which describes a generic relationship between fluctuations in the transcription rate and cell-to-cell variability in the mRNA copy-numbers. A mathematical derivation of CFT is provided and this result is used to extract interesting biological insights from three existing datasets on gene-expression systems.

Recommendation : The authors have revised their paper considerably and successfully addressed many issues that were raised on the previous version of their manuscript. In particular, the paper now includes a mathematical derivation of CFT which makes it possible to assess the generality of this relationship and properly examine the scientific contributions of the paper.

As the whole paper hinges on the CFT relationship, it is important for me to ascertain both the originality and the correctness of this relationship, before accepting the paper. In this regard, I have some issues that I request the authors to address in a subsequent revision. These issues only came to light after going through the authors' proof of the CFT in detail. I mention these issues now.

1 Major Issues

1. Basically CFT is derived from a simple model, where mRNAs *arrive* at a time-varying stochastic rate $R(t)$ and they are *serviced* (degraded) after a random time τ which is related to the survival probability according to $S(t) = \mathbb{P}(\tau > t)$. Essentially this is a infinite-server queuing model with a stochastic time-varying arrival rate $R(t)$ (see [1] for example). If we ignore the stochasticity in $R(t)$, then such a queuing model is well-studied and the exact distribution of the queue-length (or mRNA copy-number) can be computed in many cases (see Section 5.1 in [1]). In fact the formula for the mean $\langle n(t) \rangle$ that the authors derive is simply the transient version of Little's law which is well-known in queuing theory.

Once we know the variability of mRNA with time-varying (deterministic) $R(t)$, then we can add the variance contribution due to stochasticity in $R(t)$ by simply using the law of total variation. I believe this will provide a proof of CFT which is much simpler than the proof given in the supplementary material. Please explain if that is not true. In any case it would be useful to explore the queuing theory literature to find connections with CFT.

2. The formula for CFT (Equation 1) given on page 8 seems incorrect as it does not match the formula derived in the supplementary material (Equation M1-19). These two formulas will coincide if $S(t_i)$ is replaced by $S(t - t_i)$ in equation 1 for $i = 1, 2$.
3. I have some concerns with the derivation of CFT that is provided in Supplementary Method 1:
 - In deriving Equation M1-16 from M1-15 the authors take average over $\{\tau_i, \tau_j\}$. But these are deterministic dummy variables of integration and so I'm not sure what the authors mean here.
 - The definition of the Time-Correlation function (TCF) used in the derivation (Equation M1-17) is not the standard definition as the diagonal terms are ignored. This non-standard definition is not mentioned anywhere in the Main Paper which will mislead the readers. This must be clarified right after the CFT statement. Also the other specialized versions of CFT (Equations 2 and 3) seem to be derived with the standard definition of TCF. Please elaborate on this issue.
4. It must be mentioned that CFT only holds for cell-to-cell mRNA variability in a clonal population of cells, when the cells are assumed to be independent, i.e. there is no inter-cellular communication. Also the degradation machinery for mRNAs must be independent of the production machinery for CFT to hold. If you agree, please state these limitations explicitly in the introduction.

2 Minor comments

1. Page 4, paragraph 2, line 4 “ystems” should be “systems”.
2. Page 9, end of paragraph 1. Please explain why CFT cannot be derived from CME. Is it because CME only describes the evolution of probability-distribution at a single time-point and so it cannot capture temporal correlations (TCF) which are needed for the variance?

References

- [1] D. Bertsimas and G. Mourtzinou. Transient laws of non-stationary queueing systems and their applications. *Queueing Systems*, 25(1):115–155, 1997. 1

We are grateful to Reviewer 1 for his or her critical but helpful comments on our manuscript. Reviewer 1's remaining concern is that the advantages of our Chemical Fluctuation Theorem (CFT) over the existing methodologies are not fully demonstrated in our quantitative analysis of the experimental results in the previous version. In this revision, to address this important issue raised by Reviewer 1, we take advantage of our CFT to make a quantitative prediction for the dependence of mRNA noise on the mRNA lifetime distribution using the transcription model developed in this work. We also include a direct comparison between the prediction made by our CFT and accurate stochastic simulation results to confirm the correctness of the CFT's prediction. From this investigation, we find that the mRNA noise is quite sensitive not only to the mean but also to the variance in the mRNA lifetime, and that there exists a general trend in the dependence of the mRNA noise on the variance of the mRNA lifetime distribution. To the best of our knowledge, no other existing theory is capable of making this kind of quantitative prediction. Thus, by addressing the issue raised by Reviewer 1 in our manuscript, we are able to demonstrate the unique advantages of the CFT over other existing methods, and on top of this, we further demonstrate that the CFT is not only able to provide a unified, quantitative explanation of the mRNA noise for various different systems but also that the CFT can make quantitative predictions for realistic systems that have yet to be investigated, which, we believe, are significant breakthroughs in this field. Our responses to each of Reviewer 1's comments are as follows.

General comments by Reviewer 1

The revised manuscript as addressed several of my original concerns. As explained below some specific points remain unclear. However, at a general level, I am not sure if the manuscript is suitable for Nature Communications. The manuscript presents Chemical Fluctuation Theorem that is applicable to a broad class of birth-death processes (but not other forms of chemical reactions, such as binding). Then the authors apply their new theory to some datasets to provide new insight. However, the models they use can be also handled with existing methods, so the examples fail to show case the strength of the CFT (And why it is very different from other existing methods). The manuscript is also not easily accessible to a wider audience and is extremely long (150 pages of supplements). I think it may be better to publish the CFT and relevant non-trivial examples in a separate paper and then write a separate paper on the biological examples.

Response: To address this important comment, we have performed a major revision of our manuscript to more effectively emphasize and demonstrate the strength of the CFT and to show why our CFT is significantly different from other existing methods. A summary of the important changes made in response to this comment is as follows:

1) As shown in Figure 1 in the revised manuscript, we consider a more general transcription model in which the mRNA lifetime distribution is arbitrary. Accordingly, in equation (2), we present the analytic expression of the mRNA noise for the updated model. The susceptibility of the mRNA noise to the rate factor fluctuation appearing in equation (2) is not a function of the mean mRNA lifetime but a functional of the product lifetime distribution, which has not been reported elsewhere. To the best of our knowledge, the mRNA noise of this transcription model cannot be handled by any other existing method.

2) We move the text and figure associated with the quantitative analysis of the genome-wide mRNA counting statistics in *E. coli* from the main text to Supplementary Information.

3) As shown in the newly added Figure 5a-d, we use our CFT to investigate the mRNA noise for three different models of the mRNA degradation process: the sub-Poisson Michaelis-Menten enzyme process, the 1-state Poisson process, and the 2-state super-Poisson process. An exact model study of this type cannot be easily done by other existing methods. The CFT makes it clear that, when transcription is a Poisson process, the mRNA noise becomes a Poisson noise regardless of the mRNA lifetime distribution. However, through our exact model studies, we clearly show that the mRNA noise is sensitive to both the mean and fluctuation in the mRNA lifetime in the case where transcription is a non-Poisson process.

4) In the newly added Figure 5e and f, we present a quantitative prediction of our theory for the dependence of the non-Poisson mRNA noise on the mean and variance in the mRNA lifetime for the following two cases: the mRNA lifetime fluctuation caused by the non-Poisson mRNA degradation dynamics in each cell in Figure 5e and the mRNA lifetime fluctuation caused by the cell-to-cell heterogeneity in the mRNA degradation dynamics in Figure 5f. To make the prediction in Figure 5f, we generalized the CFT to encompass gene expression systems with cell-to-cell heterogeneity in the mRNA degradation dynamics. As shown in Figure 5e and f, the prediction made by the CFT and the generalized CFT is in excellent agreement with accurate stochastic simulation results.

Because no other existing model or method is capable of making this kind of quantitative prediction and explanation of the non-Poisson mRNA noise in living cells, and because our prediction for these systems can be easily compared with experiments, we believe our revised manuscript would be of great interest to experimentalists in this field as well as to theoreticians.

We also believe that the impact of this manuscript is greatest when it includes the fundamental theory, the quantitative analyses of various experiments, and the prediction for new models and experimental systems all together. The only purpose of Supplementary Information is to present the full details of our new theory and models, the mathematical methods, and the quantitative analysis of the experimental results, and to make this information more accessible to interested scholars and students. However, we can adjust the length of the Supplementary Information, upon request.

Comment 1: *In their response, the authors mention that the CFT applies more generally to feedback regulation, which is an advantage over other methods in the literature. The authors, however, do not demonstrate such an application, and the advance seems to be irrelevant to explain the experimental data. Further, we foresee several issues with the CFT in the case of feedback regulation. Specifically, for feedback regulation, the transcription rate is a potentially nonlinear function of the product number or even of its history. In this case, however, the CFT cannot provide a closed-form characterization of the noise because the correlation terms on the RHS involve higher order moments or correlation functions. Thus the CFT does not lead to a closed system of equation and therefore has similar limitations compared to other methods, e.g. Dattani, Barahona. I cannot see that the simulation methods mentioned by the authors should suffer from the same issue, however.*

Response:

We agree with Reviewer 1 that the CFT is applicable to more complex experimental systems than the systems already considered in the previous version of our manuscript, and we are currently working on an application of our CFT to the quantitative analysis of gene expression systems with feedback regulation. We feel that this on-going work would be more appropriately published in a separate manuscript, because, as Reviewer 1 notes, the length of the current manuscript is already quite extensive.

However, we do not believe that the lack of such application nullifies the advances made by the CFT for the quantitative analysis of experimental systems with feedback regulation. We believe it is an important advance in both fundamental theory and practical analysis to have discovered the general CFT, whose mathematical form is in no way affected by the presence of feedback regulation or other types of complication in the product creation process. With this knowledge, one can directly exploit the CFT to obtain a more explicit equation enabling a quantitative analysis of any given gene expression system with feedback regulation.

We also agree with Reviewer 1 in that the product creation rate is a nonlinear function of the product number for systems with feedback regulation. However, even in this case, there are a number of methods that allow one to obtain analytically tractable or numerical solutions from the CFT. In response to this comment by Reviewer 1, we have inserted one paragraph before **Summary** and present a description of a simple and general procedure to apply the CFT to a gene expression system with feedback regulation in Supplementary Note 10 in the revised manuscript.

Comment 2: *The authors also mention that the CFT applies more generally to non-exponential product lifetimes. In their derivation, they assume independent lifetimes. Non-exponential lifetimes should be observed when mRNAs compete for degradation or dependent on common factors. Non-exponential lifetimes should, therefore, be generally correlated. However, as the authors emphasise, non-exponential lifetimes are not observed experimentally! Thus it the advantage of the CFT over other methods seems irrelevant to explain the experimental data.*

Response: To address this comment, we have inserted a new section, **Prediction: mRNA noise dependency on the mRNA lifetime distribution**, before **Discussion** in our revised manuscript. It is true that the lifetime distribution of *lacZ* mRNA in *E. coli*, whose cell-to-cell variability is analyzed in the present work, are reported to be approximately an exponential lifetime distribution in refs. 44 and 66. However, in general, the mRNA degradation process is not a simple Poisson process. For example, the lifetime distribution of mRNA transcribed from *atoS*, *fabB*, and *ykgE* in *E. coli* is a super-Poisson distribution, as reported in refs. 73, 74 and 75. To address the issue raised in this comment, we demonstrate the advantage of the CFT over other existing methods by investigating the dependence of the mRNA noise on the mRNA degradation dynamics using three different models of mRNA degradation: the sub-Poisson Michaelis-Menten enzyme process, the 1-state Poisson process, and the 2-state super-Poisson process. Through these exact model studies suggest that the mRNA noise decreases with an increase in the mRNA lifetime fluctuation originating from the non-Poisson mRNA degradation dynamics.

We also present the prediction of the CFT and Model III, optimized by our analysis of the experiments, for the dependence of the non-Poisson mRNA noise on the mean and the randomness in the mRNA lifetime for the two different cases where mRNA lifetime distribution is a bi-exponential distribution. In Figure 5e, we investigate the case where the mRNA decay process in each cell is the 2-state super-Poisson process without any cell-to-cell heterogeneity. In Figure 5f, we investigate the other case where the bi-exponential mRNA lifetime distribution is contributed from two cell groups each with their own differing exponential mRNA lifetime distribution. According to the prediction, the mRNA noise increases with the cell-to-cell heterogeneity in the mRNA degradation dynamics, but interestingly, decreases as the mRNA lifetime fluctuation caused by non-Poisson mRNA degradation dynamics increases. The correctness of these predictions is confirmed against accurate stochastic simulation results.

When the mRNA degradation process is strongly coupled to the cell environment, the mRNA lifetime distribution can significantly differ from cell to cell, and the cell-to-cell variation in the mRNA lifetime distribution serves as an additional source of the mRNA noise. A simple generalization of the CFT to encompass this case is presented in Supplementary Method 8, and the prediction of the generalized CFT for this case is also demonstrated in Figure 5f.

Comment 3: *The authors point out that transcription rate fluctuations represent a small if not negligible contribution to the noise. This is in agreement with the intuition that fast oscillations in the production rate are integrated by the dynamics. The coupling term represents the biggest contribution to the noise. However, the expression for the coupling terms contains an oscillatory integrand and therefore the biggest contribution seems to stem from the integrand at zero, as shown in Fig S16. We are not convinced that this provides sufficient evidence for the existence of oscillations in the transcription rate. We wonder whether the observed oscillations could not as well be reproduced by white noise correlations of similar amplitude, at least this is what Fig S16 seems to suggest. Perhaps the fast oscillatory behaviour of the transcription rate correlation is an artefact from restricting $\Delta(x)$ to have finite support before computing the inverse Laplace transform? In summary, it has not become clear which feature of $\Delta(x)$ leads the authors to conclude on the presence of high-frequency oscillations and how this feature is represented in the data (Fig S1).*

Response: In this comment, Reviewer 1 expresses a concern about the robustness of our quantitative analysis of the experimental data shown in Figure 2 and suspects that these data can be explained not only by our model but also by a white noise model of the transcriptional rate fluctuation. However, we believe we can alleviate Reviewer 1's concern. As shown in equation (2) in the main text, the non-Poisson mRNA noise is dependent on the Laplace transform of the time correlation function (TCF) of the transcription rate, and not merely the integration of the TCF; in other words, the non-Poisson mRNA noise is sensitive to the detailed shape of the TCF. For example, as shown in Supplementary Figure S3, the model with a monotonically decaying TCF cannot explain the experimental data shown in Figure 2.

To address the issue raised in this comment, we clearly show that the experimental results cannot be quantitatively explained by the white noise model of the transcription rate fluctuation, whose TCF is Dirac's delta function, in Supplementary Figure S3 in the revised manuscript. In addition, as shown in Supplementary Figure S1 d, one can directly convert the experimental data for $\Delta(x)$ to the Laplace transform of the TCF of the transcription rate with use of equation (M3-12). A number of experimental data with negative values in Supplementary Figure S1 d can only emerge when the TCF of the transcription rate fluctuation is an oscillatory function, because the Laplace transform of a monotonically decaying function is always positive. These experimental data clearly show that the oscillatory TCF of the transcription rate fluctuation is not an artifact from representing $\Delta(x)$ as a particular mathematical function. To address this issue, in the second paragraph of **Analysis** in the main text of our revised manuscript, we explicitly mention that the experimental results shown in Figure 2 cannot be explained by assuming a model with a monotonically decaying TCF or a white noise model of the transcription rate fluctuation, referring the relevant supplementary information.

The authors would like to extend our sincere thanks to Reviewer 4 for his or her careful review and inspiring comments on our manuscript. We feel that, by addressing the important issues raised by Reviewer 4, our manuscript has been brought much closer to perfection. Our response to each of Reviewer 4's comments are as follows.

Reply to Major Comments

Comment 1:

Basically CFT is derived from a simple model, where mRNAs arrive at a time-varying stochastic rate $R(t)$ and they are serviced (degraded) after a random time τ which is related to the survival probability according to $S(t) = P(\tau > t)$. Essentially this is a infinite-server queuing model with a stochastic time-varying arrival rate $R(t)$ (see [1] for example). If we ignore the stochasticity in $R(t)$, then such a queuing model is well-studied and the exact distribution of the queue-length (or mRNA copy-number) can be computed in many cases (see Section 5.1 in [1]). In fact the formula for the mean $\langle n(t) \rangle$ that the authors derive is simply the transient version of Little's law which is well-known in queuing theory.

Once we know the variability of mRNA with time-varying (deterministic) $R(t)$, then we can add the variance contribution due to stochasticity in $R(t)$ by simply using the law of total variation. I believe this will provide a proof of CFT which is much simpler than the proof given in the supplementary material. Please explain if that is not true. In any case it would be useful to explore the queuing theory literature to find connections with CFT.

Response:

We thank Reviewer 4 for this inspiring comment. It is true that the analytic expression of the mean product number given in equation (M1-6) in Supplementary Information (SI) is the same as the transient version of Little's law in queueing theory. In our revision, after the derivation of this equation in Supplementary Method 1.2, we have added a new paragraph, explicitly mentioning that equation (M1-6) is the same as the transient version of Little's law derived by Bertsimas and Mourtzinou. However, to the best of our knowledge, the key result of this work, the Chemical Fluctuation Theorem (CFT) given in equation (1) in the main text for the variance in the product number has not been yet reported in any previous work.

In this comment, Reviewer 4 suggests an alternative derivation of the CFT, which relies on the transient version of Little's law and the law of total variance. Taking this approach, one can obtain equation (M1-27) in Supplementary Method 1 of the revised manuscript, which indeed conforms to our CFT. However, it is our belief that this derivation cannot replace our derivation of the CFT, because the time correlation function (TCF) of the product creation rate appearing in equation (M1-27) cannot be related to microscopic reaction dynamics because of how the TCF in equation (M1-27) is defined. Only through our derivation of the CFT can the definition of the TCF of the product creation rate be obtained as equation (M1-17), which enables us to relate this TCF to the microscopic reaction dynamics as demonstrated in equations (M6-11), (M6-12), or (M6-13). Conversely, in the derivation of equation (M1-27) from the law of total variance, it is not clear whether the TCF appearing in equation (M1-27) has the same definition as the TCF appearing in our CFT or how the TCF appearing in equation (M1-27) is related to the microscopic reaction dynamics. As Reviewer 4 noted in his or her third comment, the definition of the TCF, equation (M1-17), in our

derivation of the CFT does not contain the diagonal terms. However, our CFT is exact only when the TCF is defined by equation (M1-17).

As Reviewer 4 mention in his or her third comment, it is tempting to define the TCF including the diagonal terms as the standard, which is the definition given in equation (M1-28). However, were we to use equation (M1-27) with the TCF given by equation (M1-28) that includes the diagonal terms, equation (M1-27) would be incorrect, showing a great deal of deviation from the exact result, or the result of our CFT with the TCF defined by equation (M1-17). We present a clear discussion of this point in Supplementary Method 1.4 in the revised manuscript.

To address this important issue raised by Reviewer 4, we explicitly mention that the CFT is consistent with the transient version of Little's law and the law of total variance well known in queueing theory, near the end of page 9 in the revised main text, and present detailed, relevant discussion in Supplementary Method 1.4. In addition, we refer to the precise definition of the TCF of the transcription rate fluctuation and mention that only through this definition can the TCF of the product creation rate fluctuation be related to the microscopic dynamics of a reaction network model in the paragraph below equation (1).

An additional advantage of our derivation of the CFT presented in Supplementary Method 1.1-1.3 is that the derivation procedure can be extended to obtain the analytic expressions of various statistical measures including the TCF of the product number, while the law of total variance cannot be extended in such a way and only provides the variance. For these reasons, we believe our derivation of the CFT is unique and worthy of publication. In the last paragraph before **Summary** in the revised manuscript, we have explicitly mentioned this point, noting that the TCF of the protein number is crucial information required to apply the CFT to gene networks with feedback regulation.

Addressing this comment on page 9 in the main text, we refer to "D. Bertsimas and G. Mourtzinou. Transient laws of non-stationary queueing systems and their applications. Queueing Systems, 25(1):115, 1997", which was kindly cited by Reviewer 4 in this comment.

Comment 2: *The formula for CFT (Equation 1) given on page 8 seems incorrect as it does not match the formula derived in the supplementary material (Equation M1-19). These two formulas will coincide if $S(t_i)$ is replaced by $S(t - t_i)$ in equation 1 for $i = 1; 2$.*

Response:

We are grateful to Reviewer 4 for his or her careful reading of our manuscript. Equation (1) in the previous manuscript is correct only when the product creation is a stationary process. In the revised manuscript, we present the most general form of the CFT in equation (1), as Reviewer 4 suggested.

Comment 3-1: *I have some concerns with the derivation of CFT that is provided in Supplementary Method 1: In deriving Equation M1-16 from M1-15 the authors take average over $\{\tau_i, \tau_j\}$. But these are deterministic dummy variables of integration and so I'm not sure what the authors mean here.*

Response: There was a typographic error in equation (M1-15). In the revised manuscript, we clearly distinguish the deterministic dummy variables $\{t_1, t_2\}$ from the lifetime, $\tau_{i(j)}$, of the $i(j)$ -th product in equation (M1-15). The definition of $\tau_{i(j)}$ is also presented below equation (M1-15).

Comment 3-2: *The definition of the Time-Correlation function (TCF) used in the derivation (Equation M1-17) is not the standard definition as the diagonal terms are ignored. This non-standard definition is not mentioned anywhere in the Main Paper which will mislead the readers. This must be clarified right after the CFT statement. Also the other specialized versions of CFT (Equations 2 and 3) seem to be derived with the standard definition of TCF. Please elaborate on this issue.*

Response:

In this comment, Reviewer 4 raises a very important issue regarding the precise definition of the TCF of the product creation rate fluctuation. As Reviewer 4 correctly notes, the TCF $\langle R(t)R(t_0) \rangle$ defined in equation (M1-17) does not include the diagonal terms. However, it is only when we use this definition that we are able to show $\langle \delta R(t)\delta R(t_0) \rangle$ vanishes at all times for a Poisson product creation process with a constant rate, as it must vanish. In contrast, if we define the TCF by equation (M1-28), which includes the diagonal terms, the corresponding $\langle \delta R(t)\delta R(t_0) \rangle$ does not vanish, even for a Poisson product creation process, but it instead diverges when $t = t_0$. Adopting the commonly accepted notion that $\langle \delta R(t)\delta R(t_0) \rangle$ vanishes for the Poisson process with a constant rate, we believe that our definition of the TCF given in equation (M1-17) is, in fact, consistent the standard definition of the TCF of the product creation rate.

In response to this comment, we refer to the precise definition of the TCF of the product creation rate below equation (1) in the revised manuscript. In addition, in Supplementary Information 1.4., we insert four paragraphs after the derivation of equation (M1-27) from the transient version of Little's law and the law of total variance, discussing the correct microscopic definition of the TCF appearing in this equation.

Reviewer Comment 4: *It must be mentioned that CFT only holds for cell-to-cell mRNA variability in a clonal population of cells, when the cells are assumed to be independent, i.e. there is no inter-cellular communication. Also the degradation machinery for mRNAs must be independent of the production machinery for CFT to hold. If you agree, please state these limitations explicitly in the introduction.*

Response: As suggested by Reviewer 4, we more clearly define the application range of our CFT in the first paragraph on page 9 and the first paragraph on page 10 in the revised manuscript.

The correctness of our CFT is dependent on only one condition: the product decay process must be a renewal process. Under this condition, our CFT is exact, regardless of the stochastic properties of the product creation process. That is to say, the CFT holds exactly even in the presence of inter-cellular communication or other types of complication in the product creation process, which is explicitly mentioned in the paragraph before **Summary** in the revised manuscript.

In Supplementary Method 8, we present a generalization of the CFT to the case where the product degradation process is strongly heterogeneous among the cells and use this result to make a quantitative prediction for the dependence of mRNA noise on the mRNA lifetime fluctuation in Figure 5f. In Supplementary Method 1.3, we present another generalization of the CFT to encompass the case where the product lifetime distribution is dependent not only on the product survival time but also on the time at which a product molecule is created. As Reviewer 4 noted in this comment, the CFT does not exactly hold when the product degradation process is correlated with the product creation process. It is possible to generalize equation (1) into this case, however, we leave it as a topic for future research.

Response to Minor Comments

1. *Page 4, paragraph 2, line 4 “ystems” should be “systems”.*

Response: We have corrected the typographic error noted by Reviewer 4. We are grateful to Reviewer 4 for his or her meticulous reading of our manuscript.

2. *Page 9, end of paragraph 1. Please explain why CFT cannot be derived from CME. Is it because CME only describes the evolution of probability-distribution at a single time-point and so it cannot capture temporal correlations (TCF) which are needed for the variance?*

Response: Reviewer 4 makes a good point, and we now explain why the CFT cannot be derived from the chemical master equation or its variations in the second paragraph on page 9 of the revised manuscript. When the product decay process is a Poisson process, one can obtain the CFT by taking the generalized master equation approach introduced in ref. 53. However, even this approach is not applicable to the case where the product decay process is a general renewal process. This is mainly because the rate of change in the product number distribution caused by the product degradation process differs from product molecule to product molecule, except for the case where the product decay process is a simple Poisson process. We believe, for this reason, the master equation and its modern generalizations cannot provide an exact description for the product molecules with a general non-Poisson lifetime distribution.

Reviewers' comments:

Reviewer #1 (Remarks to the Author):

The authors have addressed the comments. I have no further comments.

Reviewer #4 (Remarks to the Author):

Overview: The main object of the paper is to present the *Chemical Fluctuation Theorem* (CFT) which describes a generic relationship between fluctuations in the transcription rate and cell-to-cell variability in the mRNA copy-numbers. A mathematical derivation of CFT is provided and this result is used to extract interesting biological insights from three existing datasets on gene-expression systems.

Recommendation: I sincerely thank the authors for all their effort in addressing all the issues I had raised in my last review. I strongly feel that connections with queuing theory provides more insight into CFT and its biological relevance. However I still have some doubts that I mention below. Unless these doubts are satisfactorily addressed, I won't be able to support publication of this paper in Nature Communications.

1 Major Issues

1. Despite the several explanations provided by the authors, I am still not entirely convinced that CFT cannot be derived simply by using existing queuing theory results and the law of total variation. The authors use this approach and come up with Equation M1-27 in the Supplementary Material. I believe this equation is correctly derived but I cannot understand why it does not yield the same answers as CFT (Equation M1-27 in Supplementary Material). Note that CFT and Equation M1-27 are almost identical up to the definition of the Time-Correlation function (TCF). However as these two equations provide correct formulas for the product variance and they both hold for arbitrary survival probability $S(t)$, how can the definitions of TCF be different? In general if the equality

$$\int g(x)f_1(x)dx = \int g(x)f_2(x)dx \quad (1.1)$$

holds for a large enough set of functions g , then we would expect that $f_1 \equiv f_2$.

2. I am not sure if the derivation of Equation M1-31 (Supplementary Material) from Equation M1-27 is correct. In particular if one considers the constant production rate situation, as described on page 17 of the Supplementary Material, then Equation M1-31 would yield a Fano factor of 2 (as mentioned by the authors) but Equation M1-27 should yield Fano factor 1 because

$$\langle R(\tau_2)R(\tau_1) \rangle' - \langle R(\tau_2) \rangle \langle R(\tau_1) \rangle = 0.$$

2 Minor comments

1. Supplementary Material, Page 17, line 5. "...but equation (M1-30) yields...". Do you mean equation M1-31 here?
2. Supplementary Material, Page 17, line 6. "...this result with the CTF, one can...". Here CTF should be CFT.
3. Supplementary Material, Page 17, first line of the last paragraph. "We finish this section with a simple derivation of equation (M1-32) from equation (M1-27)...". Instead of equation (M1-32) do you mean equation (M1-31) here?

We would like to thank Reviewer 4 for his or her acknowledgement of our achievement, appreciation of our effort in addressing the issues raised, and meticulous, careful reading of our manuscript. We agree with Reviewer 4 that the derivation of the Chemical Fluctuation Theorem (CFT), more specifically equations (M1-27) and (M1-28), from a transient version of Little's law and the law of total variation is correct. Otherwise, we would not have presented it in Supplementary Method 1.4.

In view of the major issues raised by Reviewer 4, our discussion below equation (M1-27) about the correct definition of the time correlation function (TCF) could have been misinterpreted as denying the correctness of equation (M1-27), which was not our intention. In the discussion we intended to show that the TCF defined in equation (M1-17) in Supplementary Method 1.3 is correct and that equation (M1-27), or the CFT derived from a transient version of Little's law and the law of total variation, would yield a correct result only if the TCF appearing in equation (M1-27) is identified as the TCF defined in equation (M1-17).

To fully address Reviewer 4's remaining concerns, in our current revision, we clearly mention that equation (M1-27) is correct and that the TCF defined in equation (M1-28) is equivalent to the TCF defined in equation (M1-17) because both derivations are exact. In addition, we have performed a major revision in our discussion below equation (M1-28) about the correct form of the TCF, addressing all the other issues raised by Reviewer 4. In the main text as well, we address this issue by revising the sentence at the third line from the bottom of page 9 as follows:

“In Supplementary Method 1, we present two different derivations of equation (1) and connect equation (1) with well-established laws in probability theory, namely, a transient version of Little's law and the law of total variation⁴⁹.”

We would like to take this opportunity to express our gratitude to Reviewer 4 for his or her truly inspiring and helpful comments throughout the review process, which has undoubtedly brought our work closer to perfection.

Please find our direct responses to each of Reviewer 4's comments below.

Comment 1:

1. *Despite the several explanations provided by the authors, I am still not entirely convinced that CFT cannot be derived simply by using existing queuing theory results and the law of total variation. The authors use this approach and come up with Equation M1-27 in the Supplementary Material. I believe this equation is correctly derived but I cannot understand why it does not yield the same answers as CFT (Equation M1-27 in Supplementary Material). Note that CFT and Equation M1-27 are almost identical up to the definition of the Time-Correlation function (TCF). However as these two equations provide correct formulas for the product variance and they both hold for arbitrary survival probability $S(t)$, how can the definitions of TCF be different? In general if the equality*

$$\int g(x)f_1(x)dx = \int g(x)f_2(x)dx$$

Holds for a large enough set of functions g , then we would expect that $f_1 = f_2$.

Response:

We agree with Reviewer 4 that equation (M1-27) with the TCF defined as equation (M1-28) is an alternative mathematical representation of the CFT given in equation (1) with the TCF defined as equation (M1-17), and we did not intend to deny this in our previous manuscript. The point we would like to make is that the simple derivation of equation (M1-27) and (M1-28) in Supplementary Method 1.4 lacks the microscopic definition of the TCF that can be naturally obtained as equation (M1-17) in the derivation presented in Supplementary Method 1.3. It is for this reason that the derivation of equation (M1-27) in Supplementary Method 1.4 cannot replace the derivation of the CFT, equation (1) with equation (M1-17), in Supplementary Method 1.3.

In response to this comment, we have revised our manuscript to present a clearer discussion about this issue below equation (M1-28).

“Since the derivation of equation (M1-27) only relies on two well-established laws, the transient version of Little’s law and the law of total variance, equation (M1-27) is exact. Note that equation (M1-27) has exactly the same mathematical structure as equation (1) or equation (M1-19). Since both equation (M1-27) and equation (1) are exact, the TCF defined in equation (M1-28) should be equal to the TCF defined in equation (M1-17), that is,

$$\langle R(\tau_2)R(\tau_1) \rangle' = \langle R(\tau_2)R(\tau) \rangle = \sum_{i=1}^{\infty} \sum_{\substack{j=1 \\ j \neq i}}^{\infty} \langle \delta(\tau_2 - t_i^c) \delta(\tau_1 - t_j^c) \rangle \quad (\text{M1-29a})$$

Comment 2:

I am not sure if the derivation of Equation M1-31 (Supplementary Material) from Equation M1-27 is correct. In particular if one considers the constant production rate situation, as described on page 17 of the Supplementary Material, then Equation M1-31 would yield a Fano factor of 2 (as mentioned by the authors) but Equation M1-27 should yield Fano factor 1 because

$$\langle R(\tau_2)R(\tau_1) \rangle' - \langle R(\tau_2) \rangle \langle R(\tau_1) \rangle = 0$$

Response:

We agree with Reviewer 4 that equation (M1-27) yields the correct result when the TCF defined in equation (M1-28) is correctly identified as $\sum_{i=1}^{\infty} \sum_{\substack{j=1 \\ j \neq i}}^{\infty} \langle \delta(\tau_2 - t_i^c) \delta(\tau_1 - t_j^c) \rangle$, the TCF defined in equation (M1-17). Equation (M1-31) in the previous manuscript was the result for the case where the TCF defined in equation (M1-28) is incorrectly identified as $\sum_{i=1}^{\infty} \sum_{j=1}^{\infty} \langle \delta(\tau_2 - t_i^c) \delta(\tau_1 - t_j^c) \rangle$ with the diagonal terms included. In our current manuscript, to convey this point as clearly as possible, we have revised the relevant discussion as follows:

“We emphasize that $\langle R(\tau_2)R(\tau_1) \rangle'$ in equation (M1-27) is different from $\sum_{i=1}^{\infty} \sum_{j=1}^{\infty} \langle \delta(\tau_2 - t_i^c) \delta(\tau_1 - t_j^c) \rangle$ although $R(t)$ is defined by equation (M1-3) or $R(t) = \sum_{i=1}^{\infty} \delta(t - t_i^c)$. Should we choose to interpret $\langle R(\tau_2)R(\tau_1) \rangle'$ by

$$\begin{aligned} & \sum_{i=1}^{\infty} \sum_{j=1}^{\infty} \langle \delta(\tau_2 - t_i^c) \delta(\tau_1 - t_j^c) \rangle \\ &= \sum_{i=1}^{\infty} \langle \delta(\tau_2 - t_i^c) \delta(\tau_1 - t_i^c) \rangle + \sum_{i=1}^{\infty} \sum_{\substack{j=1 \\ j \neq i}}^{\infty} \langle \delta(\tau_2 - t_i^c) \delta(\tau_1 - t_j^c) \rangle \\ &= \delta(\tau_2 - \tau_1) \langle R(\tau_1) \rangle + \langle R(\tau_2)R(\tau_1) \rangle \end{aligned} \tag{M1-29b}$$

equation (M1-27) would yield an incorrect result.

We can show this for the simple case where the product creation process is a Poisson process. In Supplementary Method 6, we present the relationship between the TCF of the rate fluctuation and the reaction time distribution. As shown in Supplementary Method 6, when the product creation process is a stationary renewal process with the waiting time distribution, $\psi_1(t)$, the TCF defined in equation (M1-29a) can be related to $\psi_1(t)$ by

$$\langle R(t+t_0)R(t_0) \rangle' = \langle R(t+t_0)R(t_0) \rangle = \mathfrak{Z}(t) \langle R \rangle \quad (\text{M1-30})$$

with $\hat{\mathfrak{Z}}(s) = \hat{\psi}_1(s)/[1 - \hat{\psi}_1(s)]$ (see equation (M6-13)). For a Poisson product creation process with a constant rate, R_0 , we have $\hat{\psi}_1(s) = R_0/(s + R_0)$ and $\mathfrak{Z}(t) = \langle R \rangle = R_0$ so that equation (M1-30) yields

$$\langle R(t+t_0)R(t_0) \rangle' - \langle R \rangle^2 = 0. \quad (\text{M1-31a})$$

On the other hand, if one were to mistakenly adopt equation (M1-29b) for the definition of $\langle R(t+t_0)R(t_0) \rangle'$, one would obtain a different result, namely,

$$\langle R(t+t_0)R(t_0) \rangle' - \langle R \rangle^2 = \delta(t)R_0. \quad (\text{M1-31b})$$

Between equations (M1-31a) and (M1-31b), equation (M1-31a) is obviously the correct result for a Poisson product creation process with a constant rate. It is well known that, when product creation is a Poisson process, the product number distribution is the Poisson distribution with $\langle \delta n^2(t) \rangle = \langle n(t) \rangle$. Equation (M1-27) yields the correct result only when we adopt the correct definition of $\langle R(t+t_0)R(t_0) \rangle'$ given in equation (M1-29a). This example clearly shows that equation (M1-17) or (M1-29a) is the correct definition for the TCF of the product creation rate, but equation (M1-29b) is not.”

REVIEWERS' COMMENTS:

Reviewer #4 (Remarks to the Author):

The authors have successfully addressed all my remaining concerns. I recommend publication of this paper in Nature Communications.